# MAKE A DONUT: LANGUAGE-GUIDED HIERARCHICAL EMD-SPACE PLANNING FOR ZERO-SHOT DEFORMABLE OBJECT MANIPULATION

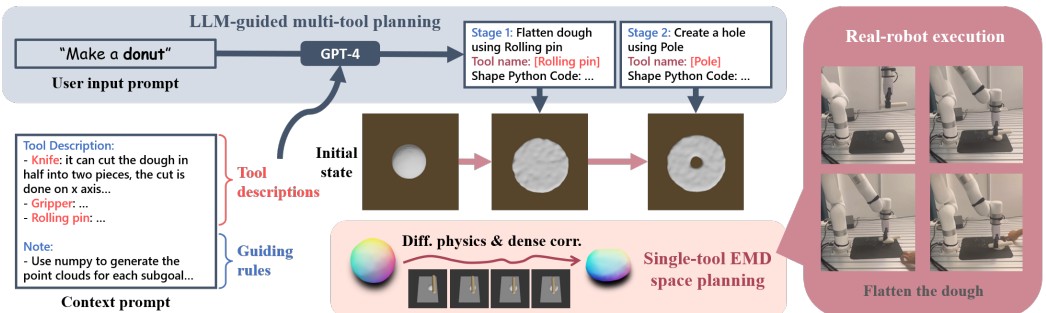

Figure 1: **Schematic illustration of our method in handling unseen dough making tasks**, where Language Models (LLMs) are utilized at a high level for task decomposition and subgoal generation, specifying tool names and generating corresponding Python code. The low-level operates on particle space controls, precisely determining the next achievable candidate iteratively without the need for prior demonstrations or task-specific training.

## ABSTRACT

Deformable object manipulation stands as one of the most captivating yet formidable challenges in robotics. While previous techniques have predominantly relied on learning latent dynamics through demonstrations, typically represented as either particles or images, there exists a pertinent limitation: acquiring suitable demonstrations, especially for long-horizon tasks, can be elusive. Moreover, basing learning entirely on demonstrations can hamper the model's ability to generalize beyond the demonstrated tasks. In this work, we introduce a demonstration-free hierarchical planning approach capable of tackling intricate long-horizon tasks without necessitating any training. We employ large language models (LLMs) to articulate a high-level, stage-by-stage plan corresponding to a specified task. For every individual stage, the LLM provides both the tool's name and the Python code to craft intermediate subgoal point clouds. With the tool and subgoal for a particular stage at our disposal, we present a granular closed-loop model predictive control strategy. This leverages Differentiable Physics with Point-to-Point correspondence (DiffPhysics-P2P) loss in the earth mover distance (EMD) space, applied iteratively. Experimental findings affirm that our technique surpasses multiple benchmarks in dough manipulation, spanning both short and long horizons. Remarkably, our model demonstrates robust generalization capabilities to novel and previously unencountered complex tasks without any preliminary demonstrations. We further substantiate our approach with experimental trials on real-world robotic platforms.

## 1 INTRODUCTION

Manipulation of deformable objects remains one of the most intricate challenges in robotics due to the inherent complexity and unpredictable behavior of such objects. Deformable objects can

be broadly categorized into two major categories: thin-shell surfaces, such as clothes Nocentini et al. (2022); Wu et al. (2019) and ropes Sundaresan et al. (2020); and volumetric objects, such as dough Lin et al. (2022c); Huang et al. (2021). In this paper, we focus on the latter and mainly study the manipulation of dough as a typical representative.

Existing works on dough-like volumetric deformable objects majorly rely on a learned dynamic model of the underlying particles Zhu et al. (2022); Arriola-Rios et al. (2020); Yin et al. (2021). However, these methods all require a substantial amount of collected or semi-auto-generated demonstrations for training the dynamic models, which poses two critical issues: firstly, the difficulty of obtaining a comprehensive set of demonstrations, particularly for long-horizon tasks; and more importantly, the limited capability of generalizing beyond the scope of the provided demonstrations.

Given this context, there is an imperative need for a more versatile and universally applicable approach to deformable object manipulation, one that can navigate the intricacies of both short and long-horizon tasks, without being overly reliant on demonstrations. This paper introduces a novel demonstration-free hierarchical planning method that addresses these challenges.

In this study, we delve into the manipulation of dough, a quintessential example of deformable object manipulation Lin et al. (2022c;b); Huang et al. (2021). As illustrated in Figure 1, our approach takes a natural language user prompt as input and leverages a large language model (LLM) to formulate a high-level plan detailing the selection of tools and the representation of intermediate subgoals at each phase. While LLMs might not produce precise low-level actions for each timestep, they exhibit proficiency in breaking down intricate tasks into manageable stages. Each of these stages exclusively involves a single tool and a piece of dough.

The concept of anchoring language to a sequential plan has been investigated in prior research Huang et al. (2022); Ahn et al. (2022); Liang et al. (2023). However, such methodologies have largely been confined to generating high-level linguistic instructions for robots for generic household tasks (e.g., "open the fridge" or "bring me the apple"). They haven't been tailored for intricate tasks like deformable object manipulation. Indeed, there is a significant gap in literature when it comes to utilizing LLMs for manipulating deformable entities, especially when the challenge entails crafting complex shapes (like donuts or baguettes) based purely on linguistic outputs. In our approach, rather than defining the robot's actions or policy linguistically at intermediate stages, we direct LLMs to express their object-centric state visualizations via Python code. This distinctive approach sets our method apart from previous techniques.

In bridging adjacent subgoal imaginations generated from LLMs, we introduce a simple but novel EMD space planning algorithm complemented by model predictive control. The algorithm evaluates the gradient of the earth mover's distance between the prevailing point cloud and the target point cloud concerning each discrete point. Subtracting this gradient from the current point cloud yields the succeeding viable candidate. This process facilitates a direct point-to-point correspondence between the existing state and the upcoming candidate, enabling the deployment of differentiable physics based on a straightforward per-point mean squared error.

Through this hierarchical strategy, our system can adeptly tackle novel, intricate tasks devoid of any demonstrations or prior training. Experimental results demonstrate that our methodology markedly enhances the efficacy of both single-tool and multiple-tool dough manipulation endeavors and can potentially transfer to real-world robotic applications.

## 2 RELATED WORKS

**Differentiable physics for deformable object manipulation.** Differentiable physics is a pivotal technique in deformable object manipulation. It exploits the gradient from differentiable simulators to derive optimal actions. Existing literature Hu et al. (2019a); Huang et al. (2021); Hu et al. (2019b); Liang et al. (2019) reveals that differentiable physics offers an efficient means to tackle short-horizon simple tasks. Nevertheless, as highlighted by Antonova et al. (2023), the reliance of differentiable physics on local gradients poses challenges. The loss landscape is often rugged with potentially spurious local optima, making it a less reliable method for certain tasks.

**Long-horizon planning for deformable object manipulation.** There's an emerging interest in long-horizon strategies for deformable object manipulation. DiffSkill Lin et al. (2022b) employs

a latent space planner to assess various skill combinations and sequences to achieve the desired outcome. Subsequently, PASTA Lin et al. (2022c) introduced the PlAnning with Spatial-Temporal Abstraction framework, melding spatial and temporal abstraction for effective long-horizon task planning. Robocraft Shi et al. (2022) advances a particle-based dynamics model using graph neural networks (GNNs) to grasp the system's structural nuances. This knowledge is then harnessed to optimize action sequences from a randomly initialized trajectory. Robocook Shi et al. (2023) adopts point cloud scene representation and leverages GNNs for modeling tool-object interactions. It then synergizes tool classification with self-supervised policy learning for crafting object manipulation plans. Nonetheless, these methodologies have their constraints. They necessitate prior insight into potential tool candidates and a predetermined number of stages for task planning, which affects their adaptability.

**Language models for robot manipulations.** Leveraging large language models for robotics is currently a bustling research domain. Recent studies such as Huang et al. (2022); Liang et al. (2023); Ahn et al. (2022) strive to dissect complex tasks into manageable sub-stages. These methods, although innovative, are primarily innocent to the underlying geometry, providing only high-level robot directives. To enrich these models with diverse modalities, SM Zeng et al. (2022) developed a modular framework where new tasks are delineated as a linguistic interaction between pre-trained models and auxiliary modules, underpinned by Socratic reasoning. PaLM-E Driess et al. (2023) engineered a versatile embodied multimodal language model tailored for a spectrum of downstream reasoning challenges. VoxPoser Huang et al. (2023) harnesses LLMs to craft voxel value maps in a 3D observation space, guiding robot-environment interactions. Since LLMs often cannot directly produce the robot's raw actions, an alternative approach is to map intricate tasks to rewards. Some other studies Goyal et al. (2019); Lin et al. (2022a) focus on curating domain-specific reward models, which necessitate abundant annotated training data. In contrast, works likeKwon et al. (2023); Yu et al. (2023) generate reward metrics automatically from pretrained LLMs, though their application is predominantly limited to rigid or articulated objects. Deformable object manipulations remain a relatively under-explored territory for LLMs, largely due to the immense degrees of freedom inherent to such tasks and the paucity of available demonstration data.

## 3 METHOD

Our method adopts a hierarchical planning approach combining both language models and low-level particle space controls. At the top level, LLMs are employed to break down a complex task into sub-stages, and output both the code to generate subgoal states and the tool name for each. We observe that LLMs obtain rich knowledge about high-level task semantics though cannot directly output raw low-level actions. At the bottom level, given the current tool and subgoal, our technique iteratively identifies the next reachable target based on the present state and subgoal. A key distinction between our approach and previous ones is that ours doesn't necessitate any demonstration or training for the target task. In Section 3.1, we detail the partitioning of complex tasks into sub-stages and the associated tools. Section 3.2 elaborates on the iterative process of determining the next goal based on the current state and subgoal.

An overview of our method is illustrated in Figure 1. Our method processes the sampled particles from the volumetric dough as input and produces the actions of the currently used tool as output. In the subsequent context, we will interchangeably use point clouds and particles.

### 3.1 MULTIPLE TOOL SELECTION AND HIERARCHICAL PLANNING

To address the challenge of coordinating between different tools in long-horizon tasks, we turn to the capabilities of large language models (LLMs), especially ChatGPT-4 OpenAI (2023). Our observations indicate that while LLMs may not precisely produce low-level actions, they excel in deconstructing intricate long-horizon tasks into several stages, each centered around a single tool. What's more, for each of these stages, the LLM can both identify the appropriate tool and generate the corresponding Python code to produce intermediate subgoal point clouds. Presented in the form of particles, these subgoals readily align with the target requirements of our proposed single-tool EMD space planning algorithm.

To guide this process, we devised a prompt template that imparts to the LLM foundational information about available tools and their potential interactions with the dough. Additionally, we introduce a set of guidelines designed to refine and direct the LLM when completing more complex, long-horizon deformable tasks. One important guideline is to force the LLM to give volume-preserving input and output at each stage, so the target is more physically realistic. In addition, we leverage the chain reasoning technique Wei et al. (2022) to help the LLM better deduce the shape parameters to satisfy this constraint. In Table 1, we provide the average relative volume change between the LLM's generated final output and the input dough for all the three evaluated tasks. Details of the prompt template can be found in Appendix A.7.

| | Donut ↓ | Baguette ↓ | TwoPancakes↓ |
|---|---|---|---|
| w/o Volume Preserving and Chain Reasoning | 73.9% | 42.5% | 65.0% |
| Ours | **9.8%** | **38.9%** | **0.0%** |

Table 1: **Volume change with and without volume-preserving and chain reasoning.**

In addition, we also ask the LLM to output the following items for each stage during planning:

- A one-line explanation of what this step is doing.
- The name of the tool to be used.
- The Python code to generate target point clouds. Do remember to add their absolute world location when generating complex shapes.
- The variable names for the input and output.
- The location of each piece in a dictionary format with a variable name as the key.
- The volume of each piece is also in a dictionary format with a variable name as the key.

Building on this, we only extract the generated Python code corresponding to each stage, from which we produce the intermediate subgoal point cloud. Other outputs, though not used, are part of chain reasoning and, therefore contribute to the final quality of the generated subgoal. The subgoal point cloud is then fed directly into our single-tool planning module. Consequently, we are equipped to tackle intricate tasks incrementally, stage by stage, eliminating the need for demonstrations.

## 3.2 SINGLE TOOL PLANNING

As the LLM can decompose a complex task into several stages with generated sub-targets, on each stage, given the current point cloud and the sub-target, we introduce a novel goal-aware planning algorithm with model predictive control.

At each step of the single-tool planning algorithm, our method initially identifies an optimal starting position for the tool. Subsequently, it forecasts the nearest attainable target and refines the actions employing differentiable physics with point-to-point correspondences, termed DiffPhysics-P2P. If this step doesn't yield progress, our model reverts the actions and re-strategizes using a new starting position. In this section, we will first talk about DiffPhysics-P2P, then the initial tool selection, and finally the failure-ware tool resetting techniques.

**DiffPhysics-P2P.** Given the current point cloud and the goal, we pinpoint the subsequent reachable point cloud by executing multiple small steps (specifically 20, as per our experiments) via gradient descent within the Earth Mover Distance (EMD) space. Formally, each step of gradient descent is:

$$\mathbf{p}_i' = \mathbf{p}_i - \alpha \cdot \frac{\partial \text{emd}(\{\mathbf{p}_i\}, \{\bar{\mathbf{p}}_i\})}{\partial \mathbf{p}_i}. \tag{1}$$

Given the current point cloud, denoted as $\mathbf{p}_i$, where $i$ is the point index, and the subgoal $\bar{\mathbf{p}}_i$, our objective is to discern the ensuing reachable candidate. This is achieved by incrementally transitioning the current point cloud towards the target within the EMD space. The candidate serves as our model's prediction of the underlying particle dynamics.

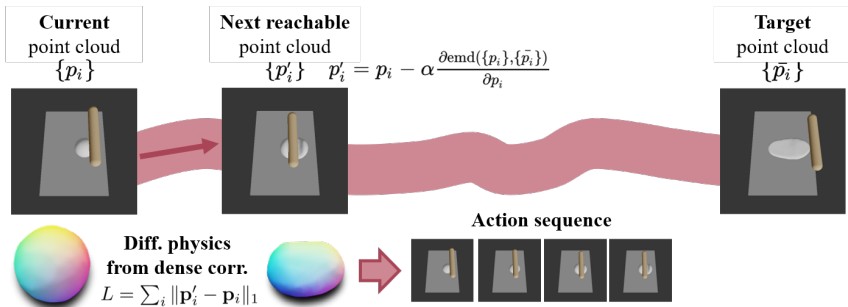

Figure 2: **EMD-space planning with DiffPhysics-P2P.** We find the next reachable target by running small steps within the EMD space. The induced point-to-point correspondence can provide better gradients when optimizing actions through differential physics.

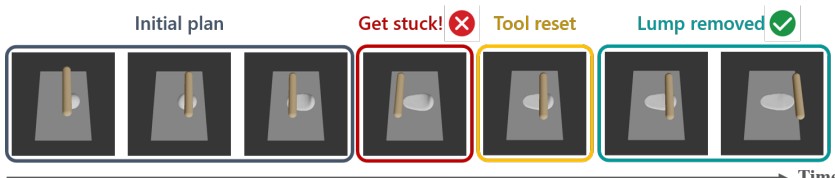

Figure 3: **Illustration on how tool reset works.** By resetting the tool position when no improvement can be made, we can jump out of the local minima and get a better global solution.

A notable advantage of the EMD gradient is its inherent capacity to furnish a one-to-one correspondence between $\mathbf{p}'_i$ and $\mathbf{p}_i$, as elucidated by Equation 1. This characteristic permits the application of the subsequent straightforward mean-squared-error loss using differentiable physics:

$$L = \sum_i \|\mathbf{p}'_i - \mathbf{p}_i\|_1. \tag{2}$$

This diverges from several preceding methodologies, wherein the naive EMD loss (expressed as $\mathrm{emd}(\mathbf{p}_i, \mathbf{p}'_i)$) is employed devoid of any point-to-point correspondence. Ablation studies underscore that our point-to-point correspondence substantially outperforms traditional differential physics by enhancing the gradient flow. An illustration of this algorithm is given in Figure 2.

**Initial position selection.** The aforementioned EMD planning algorithm demonstrates efficacy when the initial tool position is in proximity to the dough. However, challenges arise when the tool is situated at a considerable distance from the dough, resulting in the algorithm's inability to find the global minima. To figure out a good initial position for the tool, we leverage the strategy initially proposed in Li et al. (2022). Specifically, with the present deformation field deduced from the induced point-to-point correspondence (represented as $\mathbf{p}'_i - \mathbf{p}_i$), we employ the following equation to identify the candidate tool position:

$$\mathbf{x}^* = \arg\max_{\mathbf{x}} \sum_i \frac{\|\mathbf{p}'_i - \mathbf{p}_i\|_1}{\mathrm{sdf}_{\mathbf{x}}(\mathbf{p}_i) + \delta}. \tag{3}$$

The numerator encapsulates the point-to-point correspondence loss of the extant point cloud. In contrast, the denominator represents the signed distance field (SDF) of the tool when positioned at $\mathbf{x}$, evaluated at particle $\mathbf{p}_i$. From an intuitive perspective, the goal is the tool's strategic placement in close proximity to the dough, while emphasizing significant deformations within the EMD space. For practical application, we introduce a minimal margin $\delta$ in the denominator to circumvent numerical instability issues.

**Tool reset upon failures.** Integrating the previously described techniques allows us to first select an optimal initial tool position, followed by iteratively progressing towards the target as per Equation 1. However, even with an advantageous starting tool position, challenges may arise due to the

inherent intricacies of differentiable physics. As illustrated in Figure 3, consider a scenario where the task is to use a rolling pin to flatten the dough into a sphere. While initiating from a favorable position, iterating and optimizing candidates in the EMD space could land us at a local minimum. This could result in an uneven texture on one side of the dough, manifesting as a lump. To circumvent this predicament, we reset the tool's position if no advancement is observed, ensuring an escape from potential local minima. The comprehensive single-tool iterative planning process is detailed in Algorithm 1.

---

**Algorithm 1** Goal-Aware Planning with Model Predictive Control

---

**Input:** : Current system particles $\{\mathbf{p}_i\}$, target particles $\{\bar{\mathbf{p}}_i\}$
**Output:** : Predicted actions at each timestep $\{\mathbf{a}_t\}$
 1: $t := 0, need\_reset := 1, \text{emd}_{last} := \infty$
 2: **while** $t < max\_steps$ **do**
 3: $\quad \{\mathbf{p}'_i\} := \{\mathbf{p}_i\}$
 4: $\quad$ **for** $k$ **in** $1...K$ **do**
 5: $\quad\quad \{\mathbf{p}'_i\} := \{\mathbf{p}'_i\} - \alpha \cdot \nabla_{\{\mathbf{p}'_i\}} \text{emd}(\{\mathbf{p}'_i\}, \{\bar{\mathbf{p}}_i\})$ $\quad$ ▷ move particles along the grad. of emd
 6: $\quad$ **end for**
 7: $\quad$ Set $\{\mathbf{p}'_i\}$ as the next reachable candidate
 8: $\quad$ **if** $need\_reset$ **then**
 9: $\quad\quad \mathbf{x}^* = \arg\max_{\mathbf{x}} \frac{\|\mathbf{p}'_i - \mathbf{p}_i\|_1}{\text{sdf}_{\mathbf{x}}(\mathbf{p}_i) + \delta}$ $\quad\quad\quad$ ▷ find the optimal initial tool position
10: $\quad\quad$ Set initial tool position to $\mathbf{x}^*$
11: $\quad\quad need\_reset = 0$
12: $\quad$ **end if**
13: $\quad \mathbf{a}_{t:t+L} := \mathbf{0}$ $\quad\quad\quad\quad\quad\quad\quad\quad\quad\quad\quad\quad\quad\quad\quad\quad$ ▷ initialize actions for horizon $L$
14: $\quad$ **for** $j$ **in** $1...J$ **do**
15: $\quad\quad \mathbf{a}_{t:t+L} := \mathbf{a}_{t:t+L} - \nabla_{\mathbf{a}} \text{Sim}(\sum_i \|\mathbf{p}'_i - \mathbf{p}_i\|_1)$ $\quad$ ▷ do diff. physics with p2p corr.
16: $\quad$ **end for**
17: $\quad$ Execute $\mathbf{a}_{t:t+L}$ and get the particle state $\{\tilde{\mathbf{p}}_i\}$ $\quad$ ▷ $\{\tilde{\mathbf{p}}_i\}$ may be different from $\{\mathbf{p}'_i\}$
18: $\quad \text{emd}_{curr} := \text{emd}(\{\tilde{\mathbf{p}}_i\}, \{\bar{\mathbf{p}}_i\})$ $\quad\quad\quad$ ▷ calculate the actual emd after executing $\mathbf{a}_{t:t+L}$
19: $\quad$ **if** $\text{emd}_{curr} > \text{emd}_{last}$ **then** $\quad\quad\quad\quad\quad\quad\quad\quad\quad\quad$ ▷ if not making any progress
20: $\quad\quad need\_reset = 1$ $\quad\quad\quad\quad$ ▷ find another initial position to jump out of local minimum
21: $\quad\quad$ **continue**
22: $\quad$ **end if**
23: $\quad$ **if** $\text{emd}_{curr} < \tau$ **then** $\quad\quad\quad$ ▷ if we are close enough to the target do early termination
24: $\quad\quad$ **break**
25: $\quad$ **end if**
26: $\quad t = t + L, \text{emd}_{last} = \text{emd}_{curr}, \{\mathbf{p}_i\} = \{\tilde{\mathbf{p}}_i\}$ $\quad\quad\quad\quad$ ▷ run $L$ steps forward
27: **end while**

---

## 4 EXPERIMENT

In our experiments, we validate the efficacy of our method across four distinct dimensions. In Section 4.1, we deploy our hierarchical LLM-guided planning algorithm to more intricate and unseen long-horizon tasks. In Section 4.2, we conduct simpler tasks involving only one tool to authenticate the effectiveness of our single-tool planning algorithm. In Section 4.3, we perform distinct ablation studies for both single-tool and multiple-tool planning algorithms. These studies are crucial for validating the individual components of our algorithms, confirming the contribution of each component to the overall performance of the system. In Section 4.4, we translate our simulated actions to a real-world robot to manipulate the actual dough, illustrating the practical applicability and transferability of our method from sim to real.

**Baselines.** We employ the following baselines for comparisons: Firstly, we consider a simplistic gradient-based action optimization method utilizing differentiable physics, denoted as Diff. Physics. Secondly, we examine a sophisticated long-horizon planning algorithm, PASTA Lin et al. (2022c), which integrates both spatial and temporal abstraction. Thirdly, we explore a behavior cloning method that trains a goal-conditioned policy, abbreviated as BC. Lastly, we assess a model-free RL method, SAC-N An et al. (2021). For complex tasks, generating a large dataset of demonstrations is

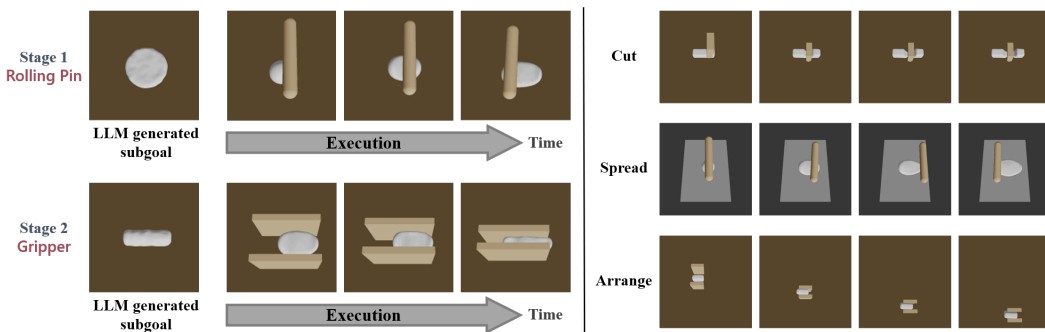

Figure 4: **Left: "Make a Baguette."** An exemplary zero-shot execution on complex long-horizon tasks. **Right: Cut, spread, arrange.** Single-tool execution results.

intractable. Thus we manually annotate a single demonstration sequence for each task for training BC and SAC-N. For PASTA, we employ its pre-trained model on single-tool demonstrations and assess its generalization capabilities on these unseen, intricate tasks. All the compared methods receive the point cloud of particles as input.

**Metrics.** We adopt the metrics in Lin et al. (2022b;c) and report the normalized decrease (score) in the Earth Mover Distance (EMD), which is computed using the Sinkhorn divergence as $\frac{\text{emd}(\{\mathbf{p}_0\},\{\mathbf{p}^*\})-\text{emd}(\{\mathbf{p}_T\},\{\mathbf{p}^*\})}{\text{emd}(\{\mathbf{p}_0\},\{\mathbf{p}^*\})}$, where $\{\mathbf{p}_0\}$ is the initial point cloud, $\{\mathbf{p}_T\}$ is the final point cloud after execution, and $\{\mathbf{p}^*\}$ is the ground-truth target 3D point cloud. Additionally, we also calculate the success rates based on pre-set thresholds (Appendix A.1).

For each multiple-tool task, we utilize the LLM's Python code output from the last stage to generate the point cloud, serving as the ground truth. We observe that these point clouds, despite being generated by the LLM, are of high quality and adeptly describe the target shape. For single-tool tasks, we follow previous literature Lin et al. (2022c;b) to sample 5 random targets at different shapes and locations. Samples of the generated target point clouds are shown in Appendix A.4. For each multiple-tool task, we generate 4 distinct responses from the LLM, with each response being assessed 5 times, culminating in a total of 20 trials per task; for each single-tool task, we follow previous literature to evaluate one trial per target, resulting in 5 trials per task.

### 4.1 MULTIPLE-TOOL SELECTION AND PLANNING

**Environment setup.** We examine three intricate long-horizon tasks that necessitate the use of multiple tools: Donut, Baguette, and TwoPancakes. As implied by their names, these tasks require the agent to create a donut, a baguette, and two pancakes, respectively. In the TwoPancakes task, the dough is initially presented as a rectangle, prepared and ready for cutting. For the Donut and Baguette tasks, the dough is initially provided in the form of a unit ball. A more comprehensive description of each task can be found in Appendix A.1.

**Results.** The quantitative results are presented in Table 2's left. It is evident from the data that our method substantially surpasses preceding approaches, exhibiting superior performance across all three tasks by a considerable margin. It is crucial to underscore that our model has never been exposed to these tasks before, and it employs the high-level stage plans generated by the LLM and the single-tool EMD space planning method to dynamically generate actions. A detailed, stage-by-stage visual representation of the process is provided in Figure 4's left, illustrating the nuanced steps and strategies employed by our method in navigating and accomplishing the tasks.

The key distinction of our approach lies in its zero-shot learning ability, which enables it to adapt to novel tasks without task-specific fine-tuning or training. This is a significant leap over the sampled-based methods, which may not provide a feasible path or require extensive data for complex shapes. The Large Language Model (LLM) plays a crucial role in our framework by charting a high-level planning path, which serves as a guide for the subsequent execution by the low-level Earth Mover's Distance (EMD) space planning algorithm. This hierarchical structure is pivotal; the LLM alone

cannot translate its generated plans into the raw actions required for robotic manipulation. Conversely, without the strategic direction provided by the LLM, the EMD space planning lacks a coherent objective, struggling to discern what end states are physically plausible for the robot to achieve. PASTA, while effective within its demonstrated scope, requires a dataset to train on in order to sample feasible intermediate states and is thus inherently limited in its ability to generalize to new shapes, e.g., donut. All their modules like VAE, cost predictor, etc., are tailored to their collected training data. This data-driven dependency hinders its application to the more complex tasks our framework successfully tackles. More qualitative comparisons are given in Appendix A.5.

| Method | Multiple-tool tasks | | | Single-tool tasks | | |
|---|---|---|---|---|---|---|
| | Donut | Baguette | TwoPancakes | Spread | Cut | Arrange |
| Diff. Physics | 0.141/0% | 0.175/20% | 0.583/0% | 0.184/20% | 0.401/60% | 0.296/20% |
| PASTA | 0.020/0% | -0.116/0% | -0.856/0% | 0.155/20% | 0.060/40% | 0.052/0% |
| BC | 0.001/0% | -0.606/0% | 0.220/0% | 0.441/60% | -0.488/20% | -0.512/0% |
| SAC-N | 0.003/0% | -0.306/0% | 0.127/0% | 0.000/0% | -2.827/0% | 0.267/0% |
| Ours | **0.346/75%** | **0.501/75%** | **0.858/65%** | **0.680/100%** | **0.685/100%** | **0.981/100%** |

Table 2: **Quantitative comparisons on both single-tool and multiple-tool tasks.**

## 4.2 SINGLE-TOOL PLANNING

**Environment setup.** In the simulation environment provided by PASTA Lin et al. (2022c), we examine three elementary tasks related to dough manipulation: Spread, Cut, and Arrange. Each of these tasks necessitates the use of only one tool at a time and can be finished within 200 time steps. More task descriptions can be found in Appendix A.1.

**Results.** Quantitative outcomes are presented in Table 2's right. It is evident that our approach consistently surpasses previous baselines by a substantial margin. The baselines fail to accomplish these tasks, whereas our method attains a 100% success rate in all of them. It is also noteworthy that, in contrast to prior learning-based approaches, our method does not necessitate any demonstration data. Remarkably, our method does not even entail any training, rendering it immediately applicable to new tasks. Qualitative results are illustrated on the right of Figure 4.

## 4.3 ABLATION STUDIES

**Multiple-tool ablations.** Figure 5 left presents our ablation studies on planning without the incorporation of high-level plans generated by the LLM. Additionally, we conduct ablation on the volume-preserving guidance with chain reasoning, which is proved to be crucial for maintaining both a high success rate and consistent target volume generation.

**Single-tool ablations** Figure 5 right presents our ablation studies on the removal of the DiffPhysics-P2P, the initial position selection component, or the tool resetting module within the single-tool planning algorithm.

## 4.4 REAL-ROBOT EXPERIMENTS

**Environment setup.** The application of this algorithm to a real-world robot is of immense interest. For this purpose, an experimental setup is established using the UFACTORY xArm 6 and some clay. Given the experimental setting and observations, the proposed planning algorithm is used to generate a trajectory in the simulator, and subsequently, the controller is employed to execute this trajectory in the real world. Despite the existence of discrepancies due to physical constraints, such as the real clay being stickier than in the simulation, the method has demonstrated its performance in generating accurate trajectories and accomplishing tasks. In Figure 6, qualitative real-robot trajectories are provided for one multiple-tool task: TwoPancakes, and two single-tool tasks: Cut, Spread.

## 5 CONCLUSIONS

We introduced a new hierarchical planning method for robotic deformable object manipulation, enabling complex tasks without prior demonstrations. This method surpasses previous demonstration-

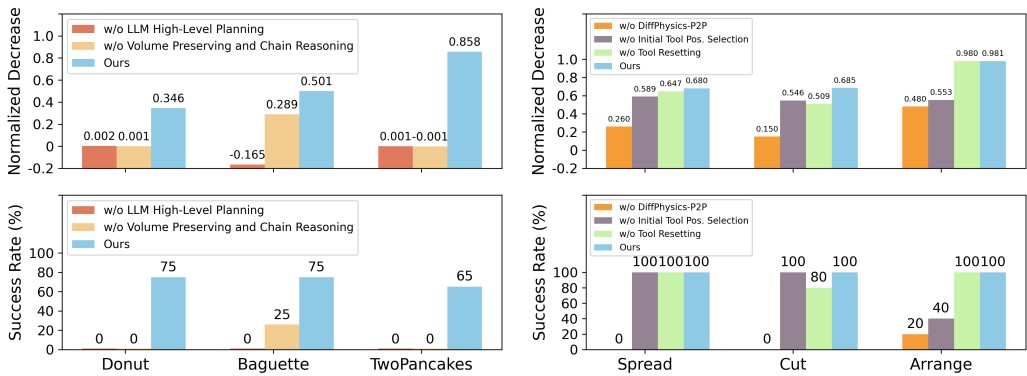

(a) Ablation results for multiple-tool experiments.    (b) Ablation results for single-tool experiments.

Figure 5: **Ablation studies.**

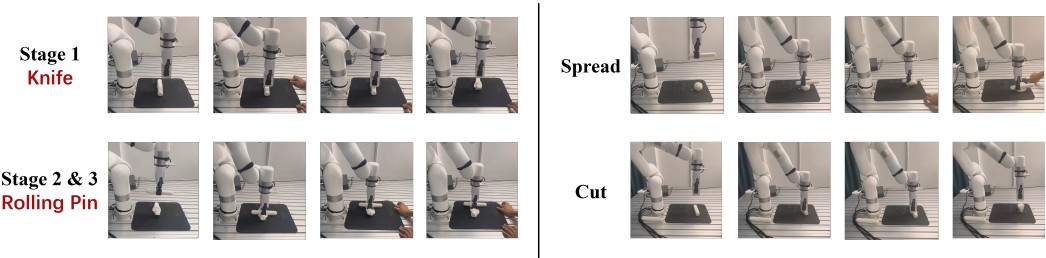

Figure 6: **Left: "Make two pancakes."** A real robot first cuts the dough and uses the rolling pin to flatten both pieces. **Right: Spread, Cut.** Single-tool action transfer results on a real-robot.

based techniques, ensuring better adaptability to new scenarios. Using large language models, it generates high-level plans and intermediate goals, which are executed through a unique closed-loop predictive control using Differentiable Physics. Our approach showed exceptional performance and adaptability in dough manipulation benchmarks, marking a significant step forward in deformable object manipulation.

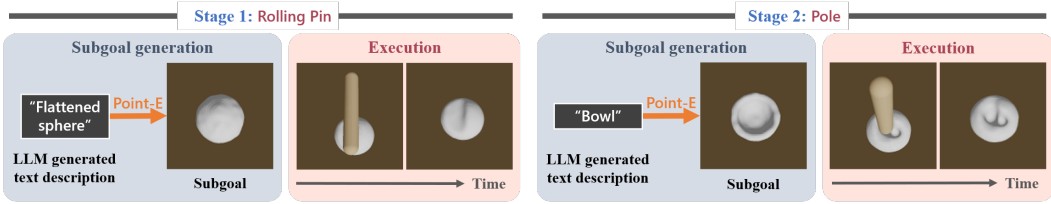

Figure 7: **"Make a bowl."** An example that leverages off-the-shelf text-to-3D algorithms to generate complex shapes.

**Limitations.**    While our method is adept at handling complex tasks involving long-horizon planning, it is limited to generating simple shapes that can be described with the Python codes produced by the LLM. Generating Python code to describe the shape of an arbitrary object is extremely challenging, if not impossible, for the LLM. However, we have made efforts to circumvent this limitation by having the LLM generate intermediate text descriptions that can be input into state-of-the-art Text-to-3D generative models, such as Point-E Nichol et al. (2022). Figure 7 illustrates an example of creating a bowl from the dough. The shape of a bowl is difficult to describe using pure Python code, so the LLM outputs text descriptions for each subgoal, like "Flattened Sphere" and "Bowl". These text descriptions are then interpreted by Point-E to generate 3D point clouds. We can then proceed with our planning algorithm, using these point clouds as our intermediate targets. As LLMs continue to evolve, we anticipate more accurate and intricate shape generation that will integrate with our proposed framework. More discussions of future works can be found in Appendix A.10.

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

## A  APPENDIX

### A.1  DETAILED DESCRIPTIONS FOR EACH TASK

**Single-Tool Tasks**

- **Spread:** Spread is a task where the dough is initially a ball. The target is to use a rolling pin to flatten it into a flat sphere. The threshold for this task is 0.4.
- **Cut:** Cut is a task where the dough is initially an elongated box. The target is to cut the dough into two pieces in the middle with a wooden knife. The threshold is 0.4.
- **Arrange:** Arrange is a task where the dough is initially a box. The target is to move the dough using a gripper to another place. The threshold is 0.7.

**Multiple-Tool Tasks**

- **Donut:** Donut is a task where the dough is initially a ball. The target is to create a donut-shaped dough. The typical way for humans to do this is to first use a rolling pin to flatten it and then create a hole with a pole in the middle. The threshold for this task is 0.3.
- **Baguette:** Baguette is a task where the dough is initially a ball. The target is to create a baguette-shaped dough. The typical way for humans to do this is to first use a rolling pin to flatten it and then compress it from both sides with a gripper. The threshold for this task is 0.5.
- **TwoPancakes:** TwoPancakes is a task where the dough is initially an elongated box. The target is to make two pancake-shaped doughs. The typical way for humans to do this is to first cut the dough in the middle, and then use the rolling pin to flatten the two pieces one by one. The threshold for this task is 0.85.

### A.2  HYPERPARAMETERS FOR SIMULATION DOUGH

We use PlasticineLab Huang et al. (2021) to simulate the dough. PlasticineLab is a Python environment that is suitable for deformable object simulation. The hyperparameters that are relevant to the properties of the dough is given in Table 3, following PASTA Lin et al. (2022c) to ensure a fair comparison.

| Parameter | Spread | Cut | Arrange | Donut | Baguette | TwoPancakes |
|---|---|---|---|---|---|---|
| Yield stress | 200 | 150 | 150 | 150 | 150 | 150 |
| Ground friction | 1.5 | 0.5 | 0.5 | 0.5 | 0.5 | 0.5 |
| Young's modulus | 5e3 | 5e3 | 5e3 | 5e3 | 5e3 | 5e3 |
| Poission's ratio | 0.15 | 0.15 | 0.15 | 0.15 | 0.15 | 0.15 |

Table 3: **Hyperparameters for dough simulation.**

Our method is also robust to a reasonable range of material properties, as shown in Table 4.

| | Donut | Baguette | TwoPancakes |
|---|---|---|---|
| Yield stress=200, Friction=1.5 | 0.365/75% | 0.523/75% | 0.864/70% |
| Yield stress=150, Friction=1.0 | 0.332/70% | 0.512/75% | 0.842/65% |
| Yield stress=150, Friction=0.5 | 0.346/75% | 0.501/75% | 0.858/65% |

Table 4: **Ablations on different choices of material hyperparameters.**

### A.3  IMPLEMENTATION DETAILS ON SIMULATED EXPERIMENTS

Our simulation environments are in line with established dough manipulation literature, notably Lin et al. Lin et al. (2022b;c). We employ both DiffTaichi Hu et al. (2019a) and PlastineLab Huang et al.

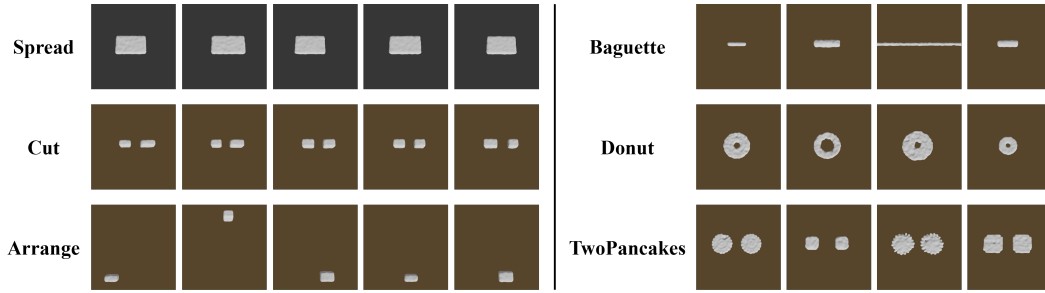

Figure 8: **LLM-generated target point clouds. Left:** Single-tool tasks. **Right:** Multi-tool tasks.

(2021) to handle differentiable physics. In all experiments, we combine three distinct losses with coefficient $(1, 1, 0.02)$ when optimizing actions with differentiable physics:

1. The point-to-point loss described in Section 4.2.

2. An SDF loss that incentivizes the tool's closeness to the dough.

3. A velocity regularization loss to ensure the dough's smooth transition.

We use a parameter $max\_steps = 200$ for single-tool tasks and for each stage of complex multiple-tool tasks. We make use of the Adam optimizer with a learning rate set at 0.02 during the differentiable physics computations. The action horizon $L$ used in our single-tool planning algorithm is 40. For the emd loss computation, our approach leans on the GeomLoss Feydy et al. (2019) library available in Pytorch. When searching for the optimal tool position, we consider the tightest axis-aligned bounding box plus a small offset above the dough depending on each tool's shape.

## A.4 TARGET POINT CLOUDS VISUALIZATION

**Single-tool tasks**  For single-tool tasks, we follow PASTA Lin et al. (2022c) to generate 5 random target dough point clouds at different places or shapes. Figure 8 left gives a detailed visualization.

**Multt-tool tasks**  For multiple-tool tasks, we utilize the LLM's Python code output from the last stage to generate the point cloud, serving as the ground truth. We observe that these point clouds, despite being generated by the LLM, are of high quality and adeptly describe the target shape. Detailed visualizations of these are illustrated in Figure 8 right.

## A.5 QUALITATIVE COMPARISON WITH INTERMEDIATE SUBGOAL VISUALIZATION

In this section, we present additional visualizations that compare our method to baseline approaches. Figure 9 depicts the execution trajectory of our method, including intermediate goals generated by the Language Learning Model (LLM). While LLM may not produce subgoals that are perfectly reachable for execution, the closed-loop execution with EMD space point-to-point planning (DiffPhysics-P2P) algorithm within our approach exhibits resilience to such imperfections. The algorithm's robustness stems from its iterative recalibration in response to ongoing observations.

Figure 10 illustrates the shortcomings of PASTA when applied to tasks involving multiple tools. PASTA struggles with tool selection during task progression and lacks generalizability in these complex scenarios.

Furthermore, Figure 11 provides a visualization of the execution process using direct EMD space planning (termed DiffPhysics-P2P) devoid of LLM-generated high-level plans. This visualization underscores the challenges inherent in the direct optimization of multiple tools within EMD space for complex tasks. Notably, the optimal transport fails to account for the underlying physics and feasibility of the actions.

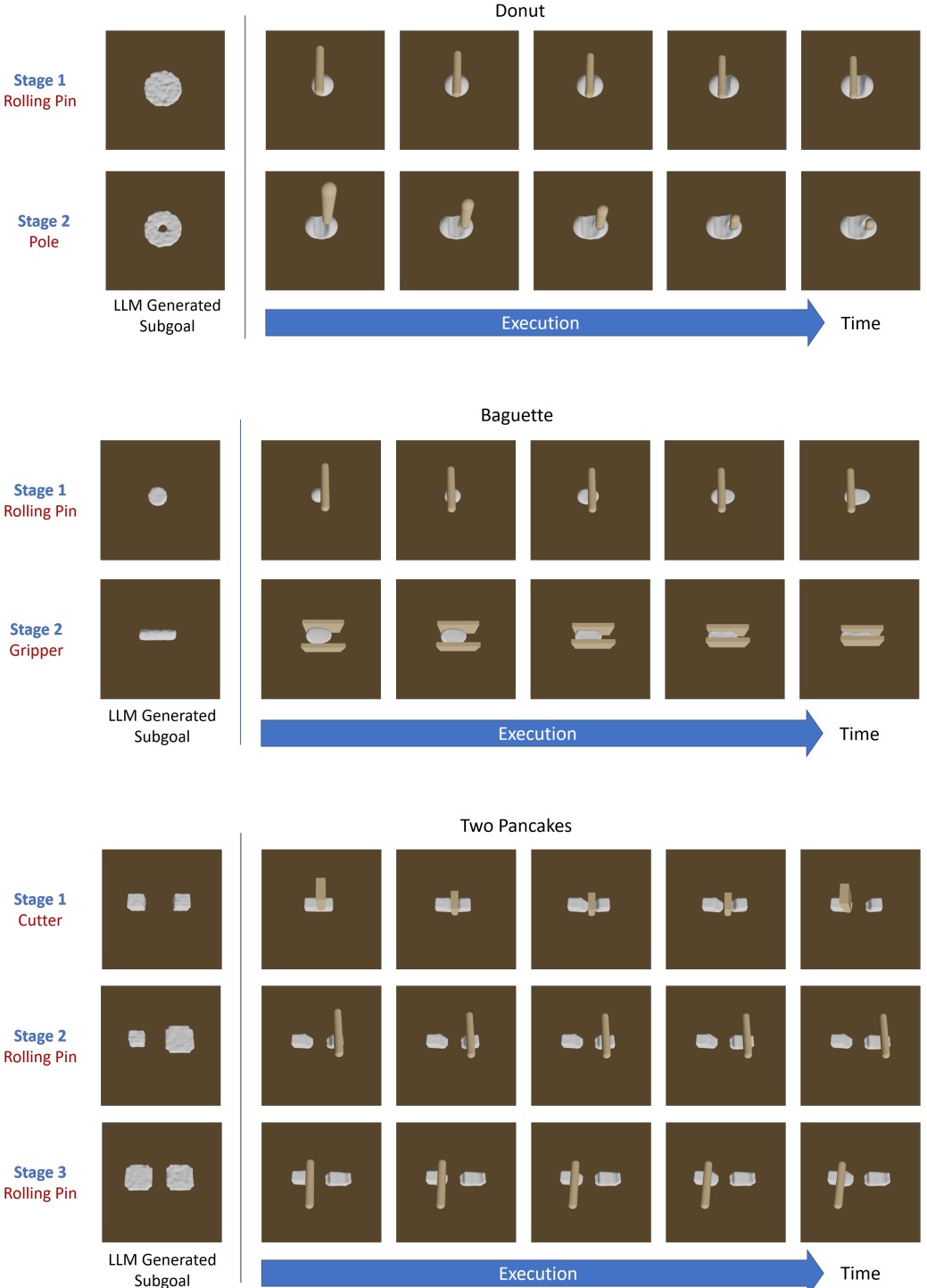

Figure 9: Execution trajectory of our method with intermediate subgoals set by the LLM.

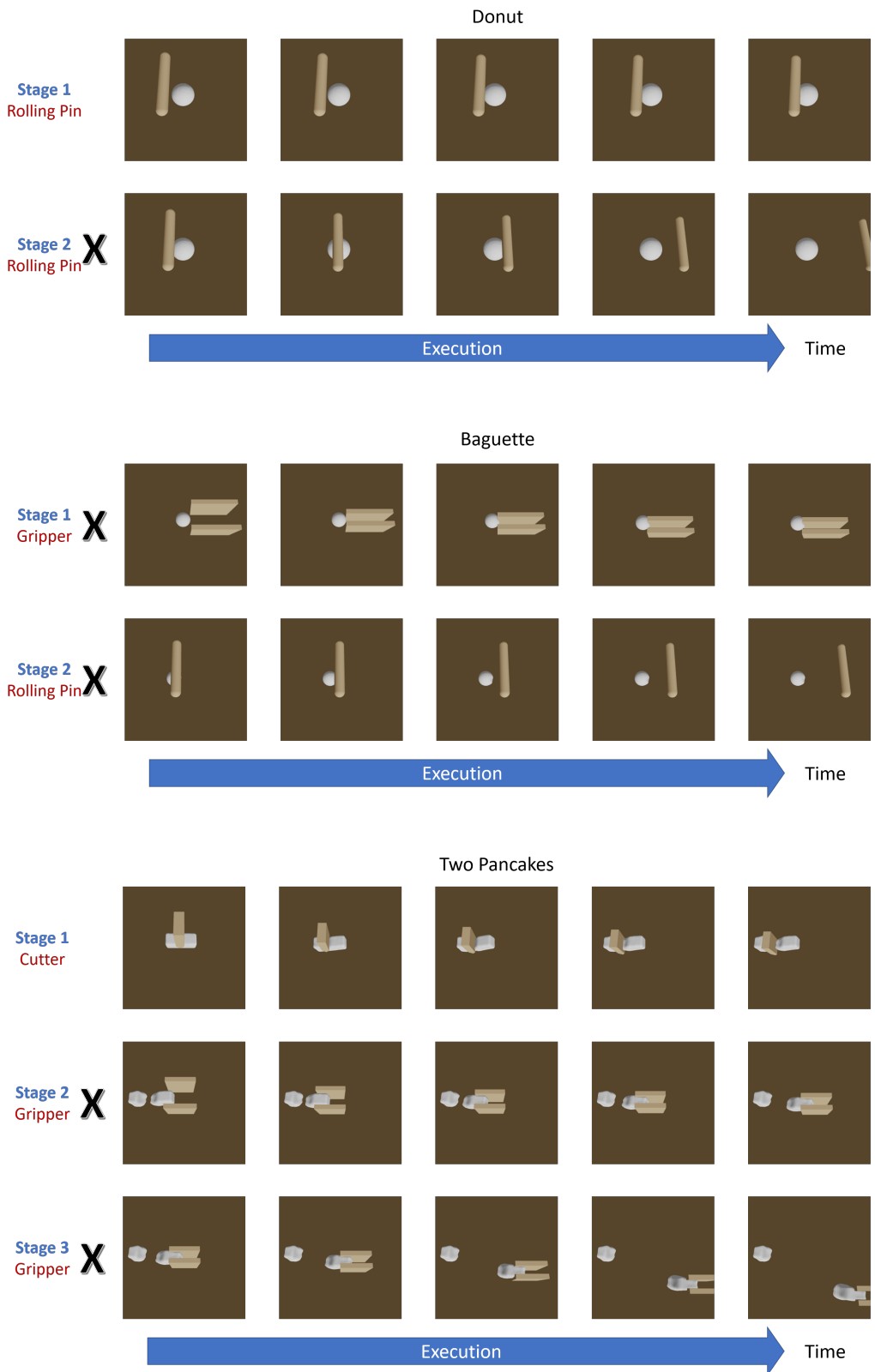

Figure 10: Visualization of PASTA's limitations in multi-tool task execution. PASTA fails to figure out the tool to use for some stages (annotated by the cross).

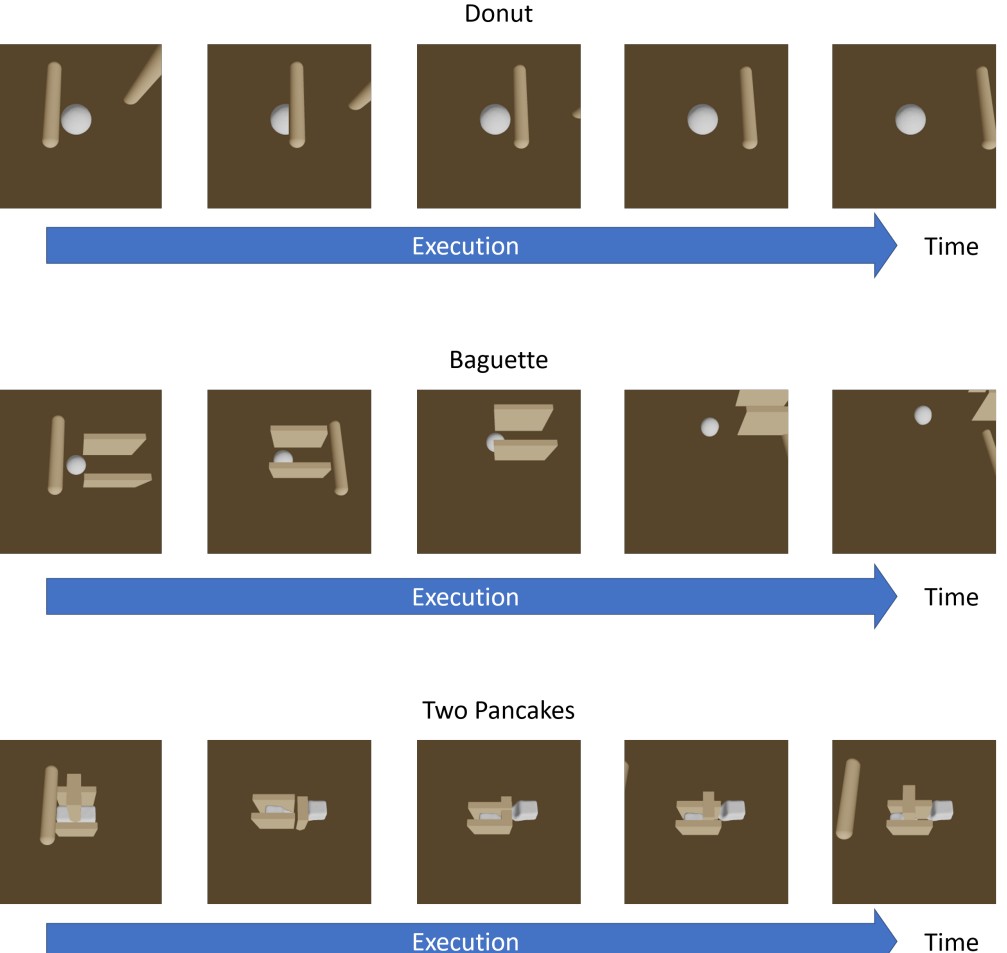

Figure 11: Execution process using EMD space planning without LLM high-level plans, demonstrating the complexity of task execution.

## A.6 NEXT REACHABLE POINT CLOUD VISUALIZATION

In this section, we visualize the topological change of next reachable point clouds throughout the process of making a donut. The visualizations demonstrate key stages, such as the intentional alteration of the dough's topology to create a hole in the middle of a donut. In this context, the intrinsic property of optimal transport (OT) that does not necessarily preserve topological structures is leveraged to our advantage. Although our tools may not execute the point cloud to perfection, the LLM-generated subgoals guide the gradient flow towards the target. The methodology allows for self-correction with each observation, ensuring that each step brings us closer to the intended outcome.

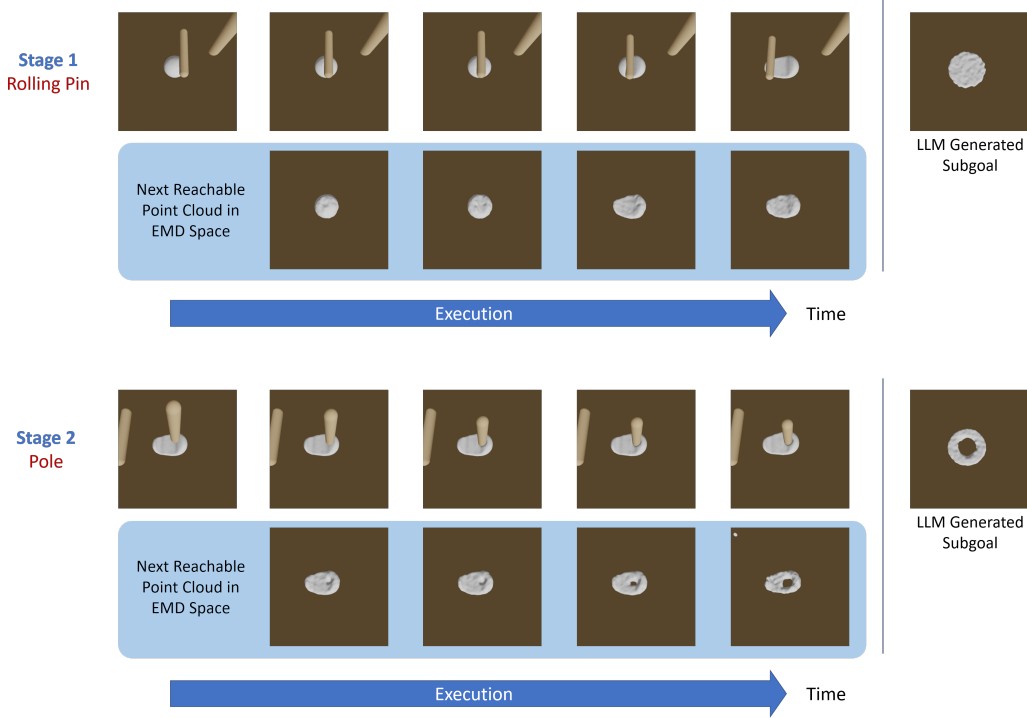

Figure 12: Next reachable point cloud visualization for making a donut. Notice that the topology is gradually changed during the second stage for making the donut. The LLM planning phase ensures that subsequent OT steps are guided towards physically realizable subgoals.

## A.7 FULL PROMPT TEMPLATE

Below is our prompt template.

I have a dough and several candidate tools, I need you to figure out which tool(s) must be used in order to reach a target.

The gravity orientation is along the y-axis.

Tool Description:

1. Pole: The pole is made of a wooden capsule. It can be used to create holes, e.g. pierce into the dough to create a hole.

2. Knife: The knife is a vertical sharp plane. It can cut the dough in half into two pieces and the cut is done on x axis.

3. Gripper: The gripper is made of two parallel planes. It can grip the dough from two opposite sides from the z-axis. The first function is to move it and the second function is to change the shape of the dough, but it cannot compress the dough from the y-axis.

4. Rolling pin: It can flatten the dough on the y-axis, e.g., from a sphere into a flat circle on the xz-plane with reduced height on the y-axis, i.e. a fat cylinder.

Note:

- Use numpy to generate the point clouds for each subgoal, in absolute world coordinates. Do not define Python functions. Define each piece of dough with a variable name. Your Python code should generate all internal points, NOT just the surface, e.g., we always want a solid ball instead of a surface sphere.

- Your response should be only a list of substeps. Start each step with the title "Step [N]: ...". You can have a step 0 to generate the initial point cloud. for each substep, you should output: 1. A one-line explanation of what this step is doing.

    2. the name of the tool to be used, in format tool_name: [tool_name]

    3. (Optional) The mathematical formula of output dimension/radius when the shape changes.

    4. The Python code to generate target point clouds. Do remember to add their absolute world location when generating complex shapes.

    5. The variable names for the input and output, in format in_var_name: [(name_in,...)], out_var_name: [(name_out,...)].

    6. The location of each piece in a dictionary format with a variable name as the key, in format loc_[name_in]: [loc], loc_[name_out]: [loc]

    7. The volume of each piece is also in a dictionary format with a variable name as the key, in format vol_[name_in]: [vol], vol_[name_out]: [vol].

- When generating the target point clouds for each subgoal, try to make it volume-preserving. When the shape changes, give a detailed mathematical formula for calculating the original volume, and how the output dimension or radius should be in order to preserve the volume. Be specific on math derivation, do not skip steps.

- Locations of dough pieces are always specified by the center of mass. When using knife for cutting, the location of each piece will move because of the split. In addition, since the knife has some thickness, each piece's center is further moved away by 0.5 unit. For example, when cutting a unit-length dough on x, each piece has length 0.5, and due to the knife's thickness, piece 1 will occupy [-1, -0.5], piece 2 will occupy [0, 0.5], with the center of mass at -0.75 and 0.75 respectively.

- Do NOT combine multiple steps, at each step we can use only one tool to manipulate only one piece of dough.

- When using Python code to generate point cloud, make sure to add their absolute location in the end.

- When using a rolling pin, try to reduce the height on the y-axis by more than 50 percent, depending on the target. Give the mathematical calculation of this height. Since the rolling pin compresses the height on the y-axis, in order to make it volume-preserving, we would expect a large radius on the xz-plane.

Example:

For example, to make a pancake from a given spherical dough, I will use a rolling pin to flatten the dough.

Task Description:

[User Prompt]

For the user prompt, we also prepend a description of the initial dough. The user prompts for the three tasks are:

- The dough is initially a round ball at (0, 0, 0) with radius 1. Make a simple donut.
- The dough is initially a round ball at (0, 0, 0) with radius 1. Make a simple baguette with its elongated side on the x-axis.
- The dough is initially a rectangle at (0, 0, 0) with dimensions (1, 0.4, 0.4). Make two pancakes.

## A.8 FAILURE CASES ANALYSIS

### A.8.1 PROMPT FAILURE

In our experiments, we find that the LLM can sometimes make it wrong when deducing the new target's shape. For example, below is one step output by the LLM in the task of making a donut.

---

**Step 1: Flatten the ball**

1. Tool: [Rolling pin].

2. Mathematical Formula:

Volume of initial ball = $(4/3)\pi r^3$

Let's say the height after flattening is $h$. The radius after flattening will be $r$. To preserve the volume: $\pi(r')^2 h = (4/3)\pi r^3$

Given $h$ is reduced by more than 50%, let's say $h = 0.4$ (40% of initial height). Solving for $r'$: $r'^2 = (4/3)r^3/0.4$, $r' = \text{sqrt}((4/3)r^3/0.4)$

3. Python Code:

```python
flattened_height = 0.4
new_radius = np.sqrt((4/3) * 1**3 / flattened_height)

def generate_cylinder(center, height, radius, points=10000):
    output = []
    for _ in range(points):
        theta = 2 * np.pi * np.random.rand()
        y = center[1] + (np.random.rand() * height - height/2)
        r = radius * np.sqrt(np.random.rand())
        x = center[0] + r * np.cos(theta)
        z = center[2] + r * np.sin(theta)
        output.append([x, y, z])
    return np.array(output)

flattened_dough = generate_cylinder((0, 0, 0), flattened_height,
    new_radius)
```

...

**Step 2: Pierce a hole in the center**

...

---

In this response, the LLM tries to use the rolling pin to reduce the height of the dough ball to $40\%$ and make a flattened cylinder as the next subgoal. However, it assumes the original height of the ball is $r = 1$, while the actual height is $2r = 2$. It fails to understand the radius is half of the full diameter. Therefore, it makes the subgoal a bit over-flattened by compressing $80\%$ of the dough in the gravity axis.

Another failure case is that in task TwoPancakes, the LLM fails to figure out the centers of the two pieces after the cutting operation. Though in the prompt we tell the LLM that the knife has some thickness and will separate the pieces by a certain offset, the LLM can sometimes give a smaller offset than the actual.

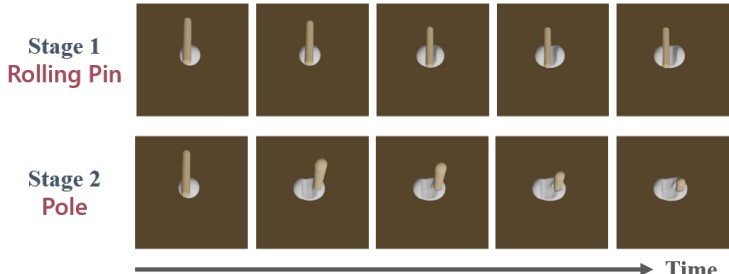

Figure 13: **An imperfect execution of making a donut.** Due to the accumulated error from different stages, the donut made by our model is not perfect.

### A.8.2    EMD-SPACE PLANNING FAILURE

In the task of crafting a donut, our planning algorithm might not yield a flawless donut, with the hole precisely at the center, due to the accumulation of errors from the preceding stage. For instance, using the prompt from the previous section as a reference, the optimal initial step would be to create a perfectly flattened cylinder using the rolling pin. However, achieving this level of precision may be practically unattainable, even for humans. Consequently, our method generates an approximately flattened cylinder with some irregularities and uneven structures in the initial stage.

In the subsequent stage, when executing planning within the EMD (Earth Mover's Distance) space, the point-to-point loss tends to be significant around the irregular, bumpy structures, as these areas exhibit the most deviation from the target. Hence, our algorithm tends to create a hole in these irregular areas instead of the center, where it ideally should be (Figure 13). This illustrates the challenges in achieving precise results due to the inherent imperfections and deviations in the initial stages of the task.

### A.9    DETAILS AND MORE VISUALIZATIONS OF REAL-WORLD EXPERIMENTS

For real-world implementations, we utilize techniques such as Poisson reconstruction with physics priors, similar to those employed in RoboCraft Shi et al. (2022), to sample particles from multi-view depth cameras. Aligning the real dough with the simulation involves using SAM Kirillov et al. (2023) to segment the dough, calculate its bounding box center, and adjust the offset to match the simulation's origin, taking into account the table plane for the z-alignment. All the tools are 3D printed to create exact replicas of the simulation.

In Figure 14, we give more visualizations of the real robot execution throughout the whole process of making two pancakes, including the intermediate subgoals generated by the LLM, and the actual outcomes achieved by the robotic execution. We show that our method can follow the guide of LLM and accomplish complex tasks with real robots.

### A.10    DISCUSSIONS OF FUTURE WORKS

**Pre-determined tool shape and size.**  Presently, the dimensions and morphology of candidate tools are pre-determined as a priori constraints, consequently limiting the range and nature of tasks that can be effectively undertaken. For instance, if we were able to use tools with the appropriate size, we can make a better donut, as shown in Figure 15. A compelling avenue for future research could be to investigate the simultaneous optimization of both the geometric shapes of these tools and the corresponding actions. By doing so, we hope that more intricate and complex tasks could be accommodated.

**Unknown physical parameters the real world.**  While our method does not require demonstrations and exhibits strong generalization to complex, unseen tasks, it does necessitate the particles and the physical parameters, such as Young's modulus, to execute differentiable physics. In real-world scenarios, such states are typically unavailable, and thus the alignment of our simulated environment with the real world using differentiable techniques presents an intriguing future direction.

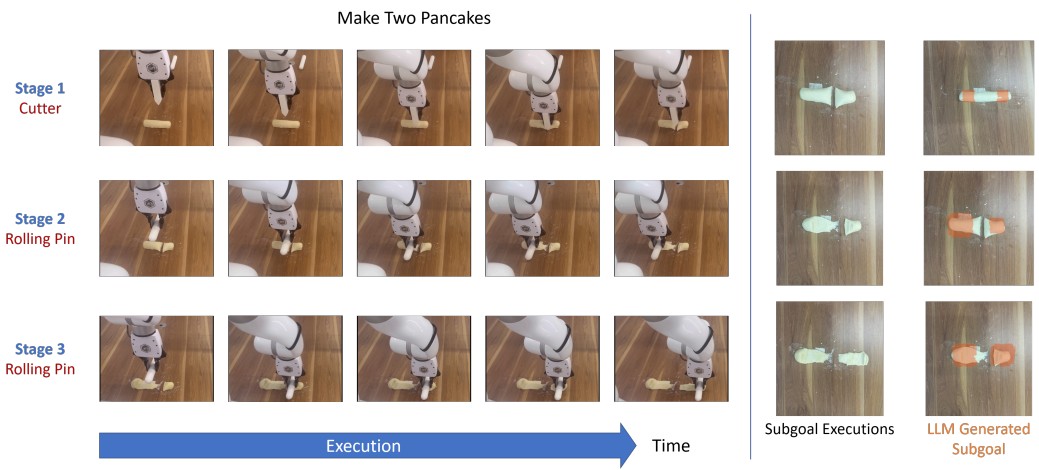

Figure 14: **Visualization of a real robot executing the task of making two pancakes.**

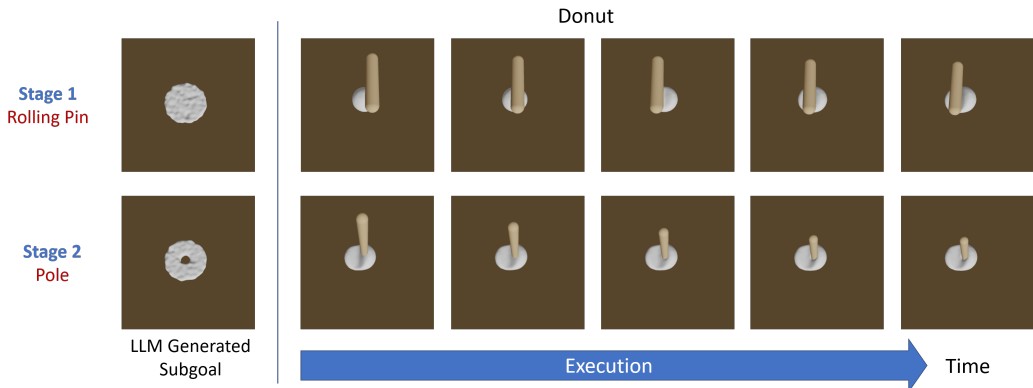

Figure 15: **We can create a better donut when using a thick rolling pin and a thinner pole.**

