# OpenReview forum: "Make a Donut: Language-Guided Hierarchical EMD-Space Planning for Zero-shot Deformable Object Manipulation"
_ICLR.cc/2024/Conference — Submitted to ICLR 2024_

### Official Review · Reviewer_ujgq · 2023-10-31

**Soundness:** 2 fair
**Presentation:** 2 fair
**Contribution:** 2 fair
**Rating:** 5
**Confidence:** 5

**Summary:**

Deformable object manipulation provides many robotics challenges. One significant challenge is the availaibility of demonstration data. The authors propose to use LLM to provide geometrical subgoals for action planning with differentiable physics optimizers to solve goal-conditioned shape planning tasks.

**Strengths:**

- **S1 - The study could offer an understanding of the advantages and limitations of LLM for robotics planning**.

As I mentioned in the weakness section as well, I encourage the author to discuss the advantages and limitations of using LLM in this setting. In its current form, the paper informs the deformable manipulation community that LLM can be used for generating subgoals.

- **S2 - The approach description is written clearly**. Although the result section could improve further, the authors provide detailed descriptions of their proposed approach.

**Weaknesses:**

- **W1 - Technical insights are unclear for the deformable and the general robotics manipulation community.**

On pre-existing tasks, the paper attempts to show that LLM can generate geometrical representations of intermediate subgoals that guide action planning.

It seems that the take-home message for the paper is that we can build a planning framework with LLM to do these pre-existing tasks. However, the results and analysis provided by the authors **do not convince me that existing frameworks such as PASTA[1] cannot accomplish the proposed tasks**.

Specifically, the result section lacks discussion, visualization, and analysis of (1) why baseline approaches like PASTA did not perform well, (2) which part of the pipeline is responsible for the failure (e.g., perception, planning, control), and (3) how do the proposed LLM planning techniques in this paper overcome these challenges.

Another suggestion I would like to give is to come up with a task that previous approaches really struggled to complete. Currently, the capability demonstrated in this paper, in both simulation and the real world, does not convince me that the proposed approach outperforms prior works.


- **W2 - Unconvincing real robot experiments.**

In the real robot experiments showcased in the supplementary video, the approach struggles to complete the desired shapes. For instance, at timestamp 01:51 of the video, the robot struggles to split the dough into halves. Plus, a human operator, in almost all tasks, needs to hold the rubber cushion below the dough to make physical interactions stable.

In addition, textual and pictorial details are missing in both the video and the paper. It would be helpful for the authors to provide **the desired final shape**, **the intermediate shapes generated by LLM**, and **the achieved shapes after real robot executions**. It would also be helpful for the authors to discuss how the incorporation of LLM makes a difference, and which part of the pipeline is responsible for failure cases.

- **W3 - The paper lacks a discussion on the limitations of optimal transport.**

The paper incorporates many techniques that hinge on optimal transport measures being faithful. However, many works such as [2] and [3] have discussed that optimal transport is limited to heavy geometrical characterization of shape differences, and lacks the physical prior to capture the "physical distance" to transform one dough shape into another. Please see Figure 4. in [2] as well as the discussion of topology preservation in [3] when directly using optimal transport gradients to transform the shape, which is what the authors are doing in this paper.

It would be helpful for the authors to provide discussions on how the proposed LLM approach could face the optimal-transport-related challenges in the limitation section.

- **W4 -  Missing visualization of the next reachable point cloud.**

The authors propose to use optimal transport gradients to geometrically transform the shape. As I mentioned in W3, this transformation does not preserve the topological structures of the initial shape and could lead to planning errors. The system therefore could struggle to generalize. It would be helpful for the authors to visualize the next reachable point cloud obtained from warping the initial shape, and how faithful can the low-level differentiable optimizer recover that shape. It would also be helpful to discuss failure cases due to incorrect next-reachable point clouds in the limitation section.


**Minor Remarks - *Incorrect characterization* of an existing work**.

> Existing works on dough-like volumetric deformable objects majorly rely on a learned dynamic model of the underlying particles Zhu et al. (2022); Arriola-Rios et al. (2020); Yin et al. (2021); **Huang et al. (2021)**.


Huang et al. (2021) does not rely on a data-driven or "learned" dynamics model. The paper proposes to leverage a built-in differentiable physics engine for trajectory optimization and is thus misrepresented in this citation. I propose that you change this to a more appropriate citation.

I thank the authors for providing the results of building the LLM-incorporated pipeline. I am happy to increase the score if the authors could make the technical insights of the paper clear, and provide more convincing results and balanced discussions.

[1] Lin et al., Planning with spatial-temporal abstraction from point clouds for deformable object manipulation, CORL 2022.

[2] Li et al., DexDeform: Dexterous Deformable Object Manipulation with Human Demonstrations and Differentiable Physics, ICLR 2023.

[3] Jean Feydy, Geometric data analysis, beyond convolutions, PhD thesis.

**Questions:**

Please see the weaknesses section above. Thanks.

---

> ### Author Response · Authors · 2023-11-15
> **Author response (Part 1 of 3)**
>
> > W1 - Technical insights are unclear for the deformable and the general robotics manipulation community.
>
> We appreciate your critical evaluation of the technical insights presented in our paper. The tasks of creating dough shapes such as donuts and baguettes are indeed not pre-existing benchmarks within deformable object manipulation; thus, they offer a novel challenge that extends beyond the capabilities of established frameworks like PASTA.
>
> The key distinction of our approach lies in its zero-shot learning ability, which enables it to adapt to novel tasks without task-specific fine-tuning or training. PASTA, while effective within its demonstrated scope, requires a dataset to train on in order to sample feasible intermediate states during planning and is thus inherently limited in its ability to generalize to new shapes, e.g., donut. All their modules like VAE, cost predictor, etc., are tailored to their collected training data. This data-driven dependency hinders its application to the more complex tasks our framework successfully tackles. We also want to point out that PASTA needs to specify the desired number of stages to execute before the planning, and we already helped them out by specifying 2 stages for making a donut, 2 stages for making a baguette and 3 for making two pancakes, but their results are still worse than ours. You can find some visualizations of PASTA in our updated A.5 (sorry to put that in the appendix due to space limit).
>
> We have now included a detailed analysis in the results section that elucidates the limitations of PASTA and similar baseline approaches. This includes a discussion on their reliance on extensive datasets and the inability to adapt to tasks involving complex manipulations such as squeezing, which our method can handle adeptly. Please see the Results paragraph of section 4.1 in our updated manuscript. We have augmented our manuscript with analyses pinpointing the failure introduced by PASTA’s inherent plan learning algorithm. This breakdown helps clarify the innovative aspects of our LLM-based planning technique.
>
> Our LLM hierarchical planning framework's strength lies in its generative capability, which allows it to conceptualize intermediate subgoals that are not only geometrically feasible but also physically plausible. This is a significant leap over the sampled-based methods, which may not provide a feasible path or require extensive data for complex shapes. The Large Language Model (LLM) plays a crucial role in our framework by charting a high-level planning path, which serves as a guide for the subsequent execution by the low-level Earth Mover's Distance (EMD) space planning algorithm. This hierarchical structure is pivotal; the LLM alone cannot translate its generated plans into the raw actions required for robotic manipulation. Conversely, without the strategic direction provided by the LLM, the EMD space planning lacks a coherent objective, struggling to discern what end states are physically plausible for the robot to achieve. This interdependence is crucial for successful task completion and is evidenced by the comparative results in Figure 5(a), Section 4.3, where the EMD space planning alone is shown to be inadequate without the LLM's high-level input.
>
> To ensure the message is clear and convincing, we have enriched the paper with the requested discussions, visualizations, and comparative analyses. This additional content will reinforce our argument that our LLM-based planning framework is indeed a significant advancement over previous methods like PASTA, particularly in its zero-shot generalization capabilities and task adaptability.
>
> > W2 - Unconvincing real robot experiments.
>
> We are grateful for your insights on the real robot experiment demonstrations. We acknowledge the instances where the robot does not perfectly execute the splitting of the dough, as highlighted at the 01:51 timestamp.
>
> The human operator's presence was solely to address the stickiness of the dough, which adhered to the rubber cushion, a non-interactive element of the environment. This was a measure taken to ensure the continuity of the experiment and in no way contributed to the task's execution, as the operator did not interact with the dough or the robot arm during the manipulation process.
>
> We have since made strides to improve our setup, including the modifications to secure the cushion in place. These improvements will be reflected in additional experiments and visualizations, which we are committed to providing in a few days and will update them in the response.

---

> > ### Author Response · Authors · 2023-11-16
> > **Author response (Part 2 of 3)**
> >
> > To ensure that our paper presents a comprehensive and transparent view of our research, we are committed to including a thorough depiction of the entire process from initial planning to execution. This will encompass detailed descriptions and visual representations of the intended final shapes, the intermediate subgoals as envisioned by the LLM, and the actual outcomes achieved by the robotic execution. This addition will be featured in the appendix section of our submission, where we have already provided visualizations for the simulated environments in Section A.5 of the updated manuscript.
> >
> > We acknowledge that there is an inherent challenge in attaining flawless execution in deformable object manipulation, a difficulty that extends to human practitioners due to the innate variability in material behavior and environmental factors. The target shapes proposed by the LLM, while geometrically precise, may not always be perfectly replicated in a physical setting. Despite this, our method demonstrates a significant capability—it consistently selects the appropriate tools and approaches to create shapes that adhere to the correct topology and general structure required for the task. This is a notable improvement over prior methodologies such as PASTA, which may not reliably select the optimal tools for each step of the process.
> >
> > > W3 - Lack of discussion on the limitations of optimal transport.
> >
> > Thank you for bringing up the critical discourse on the limitations of optimal transport (OT) in capturing the "physical distance" necessary for transforming one dough shape into another. Indeed, the works you cited correctly identify the challenges associated with using OT in isolation, particularly concerning topological preservation and its pure geometrical approach to shape transformation.
> >
> > Our framework acknowledges these limitations and proposes a novel integration of LLM with OT to address them. The LLM's role is pivotal as it injects a higher-level understanding of the task at hand, including the physical and practical considerations necessary for the manipulation of deformable objects. This is not a mere direct application of OT gradients; rather, the LLM informs the OT process by setting subgoals that are physically meaningful, which are then fine-tuned by the subsequent EMD space planning algorithm.
> >
> > Specifically, the LLM plans for subgoals that are not only geometrically reachable but also physically plausible, considering the material properties and manipulation capabilities. This is evidenced by our method's ability to create a hole in a doughnut, a task that necessitates a nuanced understanding of both the dough's physical properties and the desired end shape's topology.
> >
> > Moreover, in our ablation study presented in Figure 5(a), we demonstrate that direct application of OT is insufficient for complex, multi-step tasks and that the integration of LLM's high-level planning is critical. The LLM planning phase ensures that subsequent OT steps are guided towards physically realizable subgoals, thus mitigating the risks identified by the works you referenced.
> >
> > We hope this explanation clarifies our approach and its distinction from the direct use of OT gradients. The integration of LLM with OT is, we believe, a significant step forward in deformable object manipulation.
> >
> > > W4 - Missing visualization of the next reachable point cloud.
> >
> > In Section A.6 of our revised document, we have incorporated detailed visualizations of the reachable point flow to clearly illustrate the manipulation process. The visualizations demonstrate key stages, such as the intentional alteration of the dough's topology to create a hole in the middle of a donut. In this context, the intrinsic property of optimal transport (OT) that does not necessarily preserve topological structures is leveraged to our advantage. By manipulating the dough in a manner that does not preserve its initial topology, we can achieve the desired end shape—a task that is particularly challenging for traditional methods that strictly adhere to topological constraints.
> >
> > The next reachable point cloud is also an integral part of our method's iterative correction process. Although our tools may not execute the point cloud to perfection, the LLM-generated subgoals guide the gradient flow towards the target. The methodology allows for self-correction with each observation, ensuring that each step brings us closer to the intended outcome. We recognize that there are instances where our method may produce donuts with imperfections due to deviations in the early stages of the task, as outlined in section A.7.2 of the appendix. This limitation highlights the potential for future enhancements. With the progression of LLM models capable of incorporating precise 3D feedback and the employment of highly dexterous robotic hands, we anticipate that the execution of tasks can be refined to achieve even closer alignment with the intended outcomes.

---

> > > ### Author Response · Authors · 2023-11-16
> > > **Author response (Part 3 of 3)**
> > >
> > > > Minor Remarks - Incorrect characterization of an existing work.
> > >
> > > We appreciate your diligence in ensuring the accurate portrayal of existing work. We have moved PlasticineLab to the discussion of related works on differentiable physics.
> > >
> > > We hope that our responses have adequately addressed your concerns and provided a clearer understanding of our framework's contributions to the field. We are committed to improvement and would greatly appreciate any further feedback you may have.

---

> > > > ### Author Response · Authors · 2023-11-20
> > > > **[Updated Real Robot Experiments]**
> > > >
> > > > Dear reviewer, we are pleased to inform you that we have made major updates to our real-world experimental setup by removing the rubber cushion, where there is no human intervention now. We invite you to view the improved execution in the supplementary videos, particularly from the 1:00 timestamp onwards, which showcases the real robot execution following our planning algorithm.
> > > > Additionally, we have included image visualizations of the robot's real-world execution in section A.9 of the appendix.
> > > > We hope these updates address the concerns previously highlighted, and appreciate any feedback or further discussions.

---

> ### Comment · Reviewer_ujgq · 2023-11-21
> **Reviewer response**
>
> I deeply thank the authors for their efforts in making updates and edits. They are appreciated.
>
> > W1 & W2 - Technical insights & Real-world results.
>
> I disagree that PASTA's or other comparative approaches' results are worse, on both approach and result levels.
>
> **A. Generality and task adaptability.**
>
> On the approach level, the incorporation of LLM indeed makes everything zero-shot. I agree that there is a trend of research in our community heading in this direction and I recognize their values. However, **I wouldn't agree that learning-based approaches such as PASTA are worse or limited due to the need for task-specific training**, regardless of the amount of training data needed.
>
> In this work, I think the proposed approach **was able to circumvent policy training mostly because the control tasks in the study offer a good gradient optimization landscape for trajectory optimization**. However, it is well-known in classical control that gradient-based trajectory optimization is limiting in many ways, especially in contact-rich tasks where contact-making and breaking are not differentiable. In deformable object manipulation, the optimization landscape is especially challenging for gradient-based approaches. In these scenarios, learning-based methods such as PASTA and Diffusion Policy [1] are good solutions in my opinion.
>
> Considering the challenges the approach might face in these scenarios, it is hard for me to appreciate the technical insights of the paper from a **zero-shot and task-adaptability standpoint** which the paper advertises strongly.
>
> **B. Interpretability and quality of the results**
>
> On the results, as mentioned in the initial review, I think the technical insights are also unclear from an interpretability standpoint. **In perception**, how does the approach handle partial observation that was addressed in PASTA and DiffSkill via point cloud encoder and image encoder? For example, **how is perception performed in real-world experiments?** How are the agent states of the real world aligned with analytical simulation in the new updated video?
>
> In terms of the quality of the real-world results, PASTA's real-world results are more interpretable (in my opinion) and show overlayed subgoals in [its videos](https://sites.google.com/view/pasta-plan), and how these subgoals are achieved.
>
>
> [1] Chi et al., Diffusion Policy: Visuomotor Policy Learning via Action Diffusion, RSS 2023.
>
>
> > W3 & W4
>
> I thank the authors for their time and effort in making these updates and edits.

---

> > ### Author Response · Authors · 2023-11-21
> > **More Clarifications**
> >
> > We are grateful for your engagement and the opportunity to discuss these points further.
> >
> > > I agree that there is a trend of research in our community heading in this direction and I recognize their values. However, I wouldn't agree that learning-based approaches such as PASTA are worse or limited due to the need for task-specific training, regardless of the amount of training data needed.
> >
> > We appreciate your perspective on the role of learning-based approaches within our community. While task-specific training has its merits, we posit that the extensive data collection required for complex tasks presents a significant challenge. For instance, acquiring a comprehensive dataset for making doughnuts from scratch is not only time-intensive but may also be impractical in certain scenarios. The time and resources required to collect such data from scratch for each unique task, especially with deformable materials, are substantial and could be prohibitive.
> >
> > > In this work, I think the proposed approach was able to circumvent policy training mostly because the control tasks in the study offer a good gradient optimization landscape for trajectory optimization.
> >
> > In response to your point on policy training, our ablation studies detailed in Section 4.3 highlight the necessity of LLM for successful task completion. The LLM’s strategic high-level planning decomposes the task into manageable subgoals, making the optimization tractable. This is critical in providing a clear pathway for gradient optimization to work effectively, which is not achievable with trajectory optimization alone.
> >
> > > In perception, how does the approach handle partial observation that was addressed in PASTA and DiffSkill via point cloud encoder and image encoder? For example, how is perception performed in real-world experiments? How are the agent states of the real world aligned with analytical simulation in the new updated video?
> >
> > Addressing perception, our focus is on showcasing a planning framework rather than solving perception challenges. For real-world implementations, we utilize techniques such as Poisson reconstruction with physics priors, similar to those employed in RoboCraft, to sample particles from multi-view depth cameras. Furthermore, contrary to what might be perceived, PASTA utilizes **full** particles, similar to our approach, as evidenced in their code (https://github.com/Xingyu-Lin/PASTA/blob/main/PointFlow/pointcloud_dataset.py#L104 and https://github.com/Xingyu-Lin/PASTA/blob/main/PointFlow/pointcloud_dataset.py#L111). Aligning the physical dough with the simulation involves straightforward methods like using SAM or other detectors to segment the dough, calculate its bounding box center, and adjust the offset to match the simulation's origin, taking into account the table plane for the z-alignment.
> >
> > > In terms of the quality of the real-world results, PASTA's real-world results are more interpretable (in my opinion) and show overlayed subgoals in its videos, and how these subgoals are achieved.
> >
> > Regarding the interpretability and visualization of real-world results, we acknowledge that PASTA provides an effective presentation. However, it is important to note that they circumvent the sim-to-real gap and enhance their visualizations by employing heuristic controllers rather than the policies trained in simulation, as admitted in their limitations section: "*First, **we only transfer the planner to the real world and use heuristic controllers instead of the policy trained in simulation**. This is due to the sim2real gap caused by the differences in dough’s physical parameters, table friction, and occlusions from the robot arm*". Their approach, while yielding visually appealing results, does not fully leverage the learned actions for real-world execution. PASTA's approach in real-world applications involves transferring only the planning component of their system—specifically, decisions regarding tool selection and the portion of dough to manipulate. The execution of these plans relies on heuristic controls for each stage, rather than the direct application of their learnt policies.  Should we adopt a similar heuristic strategy, we could enhance our visualizations. Nonetheless, our aim is to present results that reflect the genuine capabilities of our planning framework, rather than relying on heuristics that may not accurately represent the learned policies' performance.
> >
> > We hope this clarification addresses your concerns and demonstrates the robustness and innovation of our proposed method.

---

> ### Author Response · Authors · 2023-11-22
> **More Clarifications (part 2)**
>
> Dear reviewer,
>
> We are very grateful for your feedback and we would like to give more clarifications in complement to our last response.
>
> > A. Generality and task adaptability.
>
> We acknowledge the significance of your point regarding the landscape of gradient optimization. It's crucial to note that even in scenarios involving contact-making and breaking, the trajectories remain differentiable, as demonstrated by frameworks such as PlasticineLab[1] and DiffTaichi[2], which effectively handle contact-rich tasks using differentiable physics. Antonova et al. [3] have highlighted the actual challenges of gradient-based trajectory optimization, particularly its susceptibility to rugged loss landscapes and local optima. In response, our method utilizes LLM to decompose complex tasks into shorter, more manageable substages. This decomposition simplifies the optimization landscape, making the trajectory optimization process smoother and more reliable, as ablated in Section 4.3. As you said, our method benefits from a favorable optimization landscape and should be taken as our advantage.
>
> [1] Huang, Zhiao, Yuanming Hu, Tao Du, Siyuan Zhou, Hao Su, Joshua B. Tenenbaum, and Chuang Gan. "Plasticinelab: A soft-body manipulation benchmark with differentiable physics." arXiv preprint arXiv:2104.03311 (2021).
>
> [2] Hu, Yuanming, Luke Anderson, Tzu-Mao Li, Qi Sun, Nathan Carr, Jonathan Ragan-Kelley, and Frédo Durand. "Difftaichi: Differentiable programming for physical simulation." arXiv preprint arXiv:1910.00935 (2019).
>
> [3] Antonova, Rika, Jingyun Yang, Krishna Murthy Jatavallabhula, and Jeannette Bohg. "Rethinking optimization with differentiable simulation from a global perspective." In Conference on Robot Learning, pp. 276-286. PMLR, 2023.
>
> Furthermore, we argue that while learning-based methods like PASTA are valuable, they inherently struggle with tasks that fall outside their training data, limiting their applicability to new, complex tasks such as doughnut making. The sim-to-real gap further exacerbates this limitation, as their learned policies often fail to transfer effectively to the real world. This is also evidenced in both visualizations and quantitative comparisons in our paper.
>
> > B. Interpretability and quality of the results
>
>  In our efforts to enhance interpretability, we have updated our supplementary materials to include videos and visualizations with overlaid subgoals. These improvements aim to provide a clearer and more intuitive understanding of how our planning framework guides the robotic manipulations towards the desired outcomes. We invite you to review these enhancements in our updated video and the visual representations found in Appendix A.9. As we explained in our previous response, PASTA uses heuristic policies (they are detailed in Section 3.1 of their appendix, one example: *The roll policy first moves the roller down to make contact with the dough. Then, based on the goal component’s length, the policy calculates the distance it needs to move the roller back and forth when making contact with the dough*.). While this can produce visually appealing results, it does not reflect the adaptive, generalized problem-solving our method aims to achieve. Our results may not look as perfect as theirs since our actions are directly derived from our optimizations rather than heuristics.
>
> We hope that these updates address your concerns comprehensively. Your feedback has been invaluable in this process, and we look forward to any additional comments you may have.

---

> ### Comment · Reviewer_ujgq · 2023-11-22
> **Reviewer response**
>
> I thank the authors for their replies and engagement in the discussion.
>
> > A. More thoughts on generality and task adaptability.
>
> The ability to break down the optimization landscape into favorable pieces could be a plus when we only consider the specific tasks presented in the study. However, since the claims of the paper focus on zero-shot and task adaptability, it is important to recognize that the low-level differentiable physics optimizer could become a significant limitation.
>
> > B. More thoughts on results in both simulation and the real world.
>
> **Simulation.** Many subgoals generated by LLM seem infeasible or not well-executed. For example, on the bottom-most row of Figure 9, the generated subgoals on the left seem to differ significantly from the results. Additionally, the donut shape seems incomplete for the second row from the top in Figure 9. I have not found a well-presented donut shape in the results, a shape championed in the title and throughout the paper.
>
> **Real world.** Thank you for clarifying the details of your perception pipeline, such as the use of SAM. I am unsure why the manuscript did not proactively include these details in the last update. I would like to encourage the authors to include these details for how real-world experiments are conducted for reproducibility, since the paper aims to speak to the robotics community.
>
> The real-world results need a lot of care to convince the robotics community. I appreciate the authors recognizing their shortcomings in real-world presentation. Starting at 01:17 in the video, the real-world shapes are heavily misaligned with the simulation / generated subgoal on the right. That is also not to mention that in the real world, a gripper is grasping on the rolling pin, which differs from the rolling pin setting the authors employ in the simulated environment. I also observed that many contacts with the dough come from the gripper instead of the rolling pin.
>
> In any case, *I deeply appreciate and recognize the authors' time and efforts in producing such real-world results during the rebuttal.*

---

> > ### Comment · Reviewer_ujgq · 2023-11-22
> > **Reviewer response**
> >
> > I recognize the authors' efforts and time during the rebuttal, bringing in new real-world results. Although I am not convinced about several parts of the paper, I still would like to increase my score from 3 to 5 to recognize the authors' time and efforts. They are appreciated- I thank the authors for the discussion.
> >
> > I hope that the authors can clarify the details of their experiments per the suggestion above, add analysis to what shapes LLM can generalize well at, and what shapes or tasks LLM struggles with.

---

> > > ### Author Response · Authors · 2023-11-22
> > >
> > > Dear reviewer,
> > > Thank you for your prompt response.
> > >
> > > >  More thoughts on generality and task adaptability.
> > >
> > > We acknowledge that our method is not universally applicable to every task, particularly those of extreme complexity. This is a point we have transparently addressed in our discussion of the limitations of our paper Nevertheless, the novelty of our contribution rests on the synergy between LLM planning and DiffPhysics-P2P, which enables us to tackle a wider array of complex tasks than previously achievable. This represents a significant step forward, pushing the boundaries of what can be realized in the realm of deformable object manipulation.
> > >
> > > > Simulation
> > >
> > > We acknowledge that our execution is not perfect and this is also quantitative evidenced in our Table 2 that we did not achieve 100% success rate. However, our method is still better than previous baselines where they immediately fail without any success (shown in the numbers of Table 2 and visualization of A.5). As we mentioned in our limitation A.10, the imperfection is partially due to the size of the tools we use. If we use a tool with an appropriate size, we can make a better donut as shown in Figure 15 of the updated paper. While current literatures (PlasticineLab, PASTA) all fix the tool to use, a compelling avenue for future research could be to investigate the simultaneous optimization of both the geometric shapes of these tools and the corresponding actions.
> > >
> > > > Real world
> > >
> > > We thanks for the suggestion and clarified the details of our experiment in A.9. All the tools are 3D printed to create exact replicas of the simulation. The only difference between sim and real is that we use grippers to grasp these tools in order to use them. In simulation we can directly change the position of rolling pin by code but in real world we cannot move the rolling pin without touching it. So we grasp the rolling pin with our gripper and since the gripper is very thin, the real-world experiment is still valid as shown in the video.

---

### Official Review · Reviewer_MTGX · 2023-11-05

**Soundness:** 3 good
**Presentation:** 3 good
**Contribution:** 2 fair
**Rating:** 6
**Confidence:** 3

**Summary:**

This paper presents a planning-based algorithm for deformable object manipulation from language instructions. Here, the deformable objects are represented as point clouds. At the core of the algorithm are 2 main components. 1. The LLM breaks down the high-level task into more concrete subtasks. 2. Within each subtask, the differentiable-physics-based planner computes the desired initial location and motion of the tool to rearrange the particles towards the desired configuration.

In the first stage, GPT-4 is queried to break down the language instruction into actionable subtasks. Concretely, GPT-4 is provided with the available tools, the initial point cloud representation of the dough, and the task. It is then queried to select the appropriate tool and write a Python script to generate the desired ending configuration for each subtask.

In the planning stage, the planner takes the assigned tool, the current dough configuration, and the desired configuration. It iteratively plans to make progress towards the desired point cloud. First, each particle is shifted toward the direction that reduces the EMD between the current and goal point cloud, creating an intermediate goal with dense p2p correspondence. Next, the tool is placed closest to the current point cloud. Then, the tool's movement is optimized by taking gradient steps with the differentiable physics simulator to morph the point cloud toward the intermediate goal.

The authors compared the proposed method with an optimization-based method, a planning method, behavior cloning, and reinforcement learning in 6 tasks in simulation. The results demonstrate that the proposed method achieves significantly higher performance. The method is also successfully deployed on a real robot with planning done in simulation, which shows robustness in modeling discrepancy.

**Strengths:**

- The problem that this work addresses is a relevant challenge in robotic manipulation.
- The formulation and the main algorithm are clearly communicated.
- The experiment task settings are clear. The ablation study results are persuasive.
- The algorithm is demonstrated on real hardware.
- The authors provide insightful analysis of some of the failure cases.

**Weaknesses:**

- The solution proposed in this work is heavily tailored towards the deformable object manipulation setting. It also requires the point-cloud representation and a differentiable physics simulator. This limits the range of applications of the method.
- The LLM's plan is open-loop. In this case, the next stage assumes the first stage is feasible. When the planner fails to complete a stage, the next stage starts at a bad initial state.
- The input to the LLM is limited to the language description of the dough. If the LLM can get a more accurate/grounded idea of the dough configuration, it might be able to provide higher-quality plans.
- The baseline methods are not sufficiently explained. (See questions)

**Questions:**

- Can you provide more details on the baseline methods used in the simulated experiments? For example, what is the observation space for the BC and SAC agents?
- Maybe I missed something, but why is the numerator needed in equation 3? It appears to me that neither $\mathbf{p}_i$ nor $\mathbf{p}'_i$ is a function of $\mathbf{x}$.

---

> ### Author Response · Authors · 2023-11-15
> **Author response**
>
> > The solution proposed in this work is heavily tailored towards the deformable object manipulation setting. It also requires the point-cloud representation and a differentiable physics simulator. This limits the range of applications of the method.
>
> We acknowledge your observation regarding the specificity of our framework towards deformable object manipulation. This focus is intentional, as deformable objects present a unique set of challenges distinct from rigid body manipulation. The paper aims to advance the field within this defined scope, contributing a versatile planning framework tailored to the complexities of deformable materials. The utilization of point-cloud representations and a differentiable physics simulator aligns with contemporary methodologies in the field[1][2][3], ensuring that our approach is both relevant and comparable to existing research.
>
> [1]Li, S., Huang, Z., Chen, T., Du, T., Su, H., Tenenbaum, J. B., & Gan, C. (2023). DexDeform: Dexterous Deformable Object Manipulation with Human Demonstrations and Differentiable Physics. arXiv preprint arXiv:2304.03223.
>
> [2]Li, S., Huang, Z., Du, T., Su, H., Tenenbaum, J. B., & Gan, C. (2022). Contact points discovery for soft-body manipulations with differentiable physics. arXiv preprint arXiv:2205.02835.
>
> [3] Xingyu Lin, Carl Qi, Yunchu Zhang, Zhiao Huang, Katerina Fragkiadaki, Yunzhu Li, Chuang Gan, and David Held. Planning with spatial-temporal abstraction from point clouds for deformable object manipulation. In Conference on Robot Learning (CoRL), 2022c.
>
>
> > The LLM's plan is open-loop. In this case, the next stage assumes the first stage is feasible. When the planner fails to complete a stage, the next stage starts at a bad initial state.
>
> The planning framework we propose is inherently flexible and does not rely on the success of previous stages, as each subgoal generated by the LLM is independent. The online EMD space planning algorithm is robust to variations in initial states and is capable of directing the manipulation towards the desired subgoal at each stage though not perfect. We also provide visualizations on intermediate subgoals and the next reachable point cloud at each timestep to reach the subgoal, in the updated A.5, A.6. We acknowledge the complexity of implementing a closed-loop system for LLM that involves real-time perception and reconstruction. While we recognize this as an important direction for future research, our current work is focused on the planning aspect within the constraints of available technology.
>
> > The input to the LLM is limited to the language description of the dough. If the LLM can get a more accurate/grounded idea of the dough configuration, it might be able to provide higher-quality plans.
>
> Your point about the LLM's input is well-taken. Indeed, a more nuanced understanding of the dough's configuration could potentially enhance plan quality. Our framework is built to be compatible with advancements in perception technologies. We aim to address these perceptual challenges in future work, as our current focus is to establish a robust planning strategy within the confines of present-day perception capabilities.
>
> > Can you provide more details on the baseline methods used in the simulated experiments? For example, what is the observation space for the BC and SAC agents?
>
> In our experiments, all baseline methods, including BC and SAC agents, utilize the point cloud of the dough as their observation space, plus the current translation and rotation of the tools. This standardization ensures a fair and consistent comparison across different methods. We use a PointNet-like backbone to encode the input point cloud with embedding dim set to 128. For SAC, the learning rates for actor and critic are set to 3e-4, with the reward decaying ratio $\gamma$ set to 0.99.
>
> > Maybe I missed something, but why is the numerator needed in equation 3?
>
> Thank you for your question regarding the numerator in Equation 3. Upon reviewing the equation, we have realized that there was an omission in the manuscript. We apologize for this oversight. The equation indeed requires a summation symbol to accurately represent the aggregation of the terms across all indices \( i \).
> The corrected form of Equation 3 should be:
> $ \mathbf{x}^* = \arg\max_\mathbf{x} \sum_i \frac{\|\mathbf{p}_i' - \mathbf{p}_i\|_1} { \mathrm{sdf}_x(\mathbf{p}_i) + \delta}.$
> It depends on the numerator if we write out all the terms and find a common denominator.
>
> We hope this clarifies your concerns and we appreciate any further feedback you may have.

---

> ### Author Response · Authors · 2023-11-22
>
> We are grateful for the valuable feedback you have provided on our work thus far. Your insights have been instrumental in helping us refine our approach and presentation. As the deadline for discussion is approaching, we would greatly appreciate it if you could provide any additional questions, comments, or concerns at your earliest convenience.

---

> > ### Comment · Reviewer_MTGX · 2023-11-23
> >
> > Thank you for the comprehensive responses and the attention given to addressing the issues highlighted in my initial review.
> >
> > Upon reviewing additional feedback from my peers, it has become apparent that the subject of deformable object manipulation encompasses greater complexity than initially considered. Despite this, my assessment remains unchanged, as I believe the system presented is pretty complete for simpler tasks.

---

### Official Review · Reviewer_Hik8 · 2023-11-08

**Soundness:** 3 good
**Presentation:** 3 good
**Contribution:** 3 good
**Rating:** 5
**Confidence:** 4

**Summary:**

The paper presents a hierarchical planning approach for dough manipulation. It uses LLMs to generate high-level plans and subgoals, while employing closed-loop model predictive control at the low-level with physics-based strategies.

**Strengths:**

- The proposed approach does not require demonstrations and uses LLMs to decompose high-level tasks and generate subgoals.
- The low-level planning operates on particle space controls and uses DiffPhysics-P2P loss in the EMD space.
- Experimental results show the effectiveness and generalization capabilities of the technique in dough manipulation tasks. The approach is further validated through experimental trials on a real-world robot.

**Weaknesses:**

- It appears that the LLM does not undergo any interactions or corrections after generating the point clouds for each stage.
- The method described is effective for complex tasks with long-term planning, but it can only generate simple shapes using Python code.
- Error recovery seems incomplete, and there still appears to be a risk of getting stuck after resetting the tool. There is no comprehensive mechanism for early termination.

**Questions:**

- Is the human effort required to develop these prompts more costly than define traditional task schema?
- The paper includes the hyperparameters used for dough simulation in A2, but it lacks analysis or discussion on the selection process and their impact on the results. Given that the physical properties of the dough can vary, how should this issue be addressed to make it more useful for different setups?
- You extract the generated Python code from each stage to generate intermediate target point clouds. How to ensure that this generated point clouds are plausible for the real world setting, and how to align them?
- How to use parallel grippers to manipulate rolling pins and other tools?

---

> ### Author Response · Authors · 2023-11-15
> **Author response**
>
> > It appears that the LLM does not undergo any interactions or corrections after generating the point clouds for each stage.
>
> We appreciate your insight on the role of LLM in our framework. The LLM indeed provides a high-level planning blueprint. Our EMD space planning algorithm is designed to be responsive and adaptive, using the LLM's output as a guide while continuously refining the approach based on real-time observations. This dynamic interplay ensures that potential discrepancies between the subgoals and the physical execution are adjusted for at each step. You can also find more visualizations in our updated A.5 and A.6.
>
> > The method described is effective for complex tasks with long-term planning, but it can only generate simple shapes using Python code.
>
> Your observation regarding shape complexity is astute. While our current system does indeed focus on simpler shapes, the framework's core contribution lies in its general approach to integrating LLM with spatial planning. The complexity of generated shapes is acknowledged as a limitation and an avenue for future research. Our focus remains on establishing a robust planning methodology, which can later be enhanced to tackle more complex shapes as text-to-3D technologies advance.
>
>
> > Error recovery seems incomplete, and there still appears to be a risk of getting stuck after resetting the tool. There is no comprehensive mechanism for early termination.
>
> Our framework incorporates a continuous correction mechanism within the EMD space planning algorithm, which actively seeks to reconcile deviations from the intended subgoal after each action, which can be further validated by the provided visualizations in A.5 and A.6. We recognize the importance of an early termination condition and have introduced a termination criterion within our algorithm, as detailed in L23 of Algorithm 1 of the revised manuscript, to prevent potential deadlocks and ensure efficient task completion.
>
> > Is the human effort required to develop these prompts more costly than defining a traditional task schema?
>
> The effort required to develop prompt templates is minimal when compared to the traditional task schema definition. The use of LLM for planning is grounded in its ability to interpret and act on the context provided by the template. This is a one-time effort, after which users can interact with the system using simple, intuitive prompts. This significantly reduces the cognitive load and technical expertise required from the user.
>
> > The paper includes the hyperparameters used for dough simulation in A2, but it lacks analysis or discussion on the selection process and their impact on the results. Given that the physical properties of the dough can vary, how should this issue be addressed to make it more useful for different setups?
>
> The selection of hyperparameters for dough simulation is consistent with established benchmarks, specifically the PASTA framework, to ensure a fair comparison. We also conduct an experiment to demonstrate the robustness of our approach across a reasonable range of material properties.
> |            | Donut      | Baguette   | TwoPancakes |
> |:---------- |:----------:|:----------:|:-----------:|
> | Yield stress=200, Friction=1.5 | 0.365/75%  | 0.523/75%  | 0.864/70%   |
> | Yield stress=150, Friction=1.0 | 0.332/70%  | 0.512/75%  | 0.842/65%   |
> | Yield stress=150, Friction=0.5 | 0.346/75%  | 0.501/75%  | 0.858/65%   |
> We agree that extending this robustness to a wider array of materials presents an interesting challenge for future work, which would involve teaching the LLM to interpret and adapt to varying physical dynamics.
>
> > Q3. How to ensure the intermediate goal is feasible; how to align it with the real world?
>
> Our experimental evidence supports the feasibility of LLM-generated intermediate goals for common bakery items like donuts, baguettes, and pancakes. While there may be limits to the complexity of shapes the current tools can produce, we believe these are reasonable constraints given the scope of this paper. Aligning these goals with the real world is achieved by standardizing the initial frame of reference between the LLM coordinate system and the physical environment.
>
> > Q4. How to use parallel grippers to manipulate rolling pins and other tools?
>
> This objective can be attained by installing a grasp-compatible structure on the tools or by adjusting the gripper's opening distance to securely accommodate the tool dimensions. Our configuration is designed to ensure the gripper makes optimal contact with the tools, whether it be clasping the midpoint of a rolling pin or the apex of a cutter. These adjustments are rudimentary and not the primary focus of our research.
>
> We hope that our responses address your queries, and we appreciate any further discussion on our work.

---

> ### Author Response · Authors · 2023-11-22
>
> We are grateful for the valuable feedback you have provided on our work thus far. Your insights have been instrumental in helping us refine our approach and presentation. As the deadline for discussion is approaching, we would greatly appreciate it if you could provide any additional questions, comments, or concerns at your earliest convenience.

---

> > ### Comment · Reviewer_Hik8 · 2023-11-23
> >
> > Thank you for your feedback. I appreciate the innovative tasks you are working on. I will continue to follow up on your future work in unleashing the potential of LLM in robotics applications.

---

> > > ### Author Response · Authors · 2023-11-23
> > >
> > > We appreciate your acknowledgment of our response and are glad to hear you liked our work. We have strived to address all the points raised comprehensively. If you feel the revisions and clarifications have enhanced the value of our work, we hope this might be reflected in your assessment. Thank you for your thoughtful consideration.

---

### Official Review · Reviewer_c5qh · 2023-11-08

**Soundness:** 2 fair
**Presentation:** 3 good
**Contribution:** 2 fair
**Rating:** 5
**Confidence:** 3

**Summary:**

The paper studies the problem of deformable object manipulation planning. The authors propose leveraging LLM for generating tool names as well as Python code for producing intermediate point cloud subgoals. Then the framework utilizes model predictive control with differentiable simulation to derive the manipulation strategies. Experiments are performed in both simulation and real-world.

**Strengths:**

The problem of deformable object manipulation is important, and the proposed method of using LLM for generating intermediate point cloud subgoals is novel and interesting;

The results look good from the evaluation perspective, and the real-world experiments further demonstrate the applicability of the work.

**Weaknesses:**

The subgoal point cloud is created using the Python code generated by the LLM, with constraints on the shape's complexity or intricacy;

This method necessitates users to submit comprehensive prompts, and the system's effectiveness is closely tied to the quality of the prompt data provided;

The system appears to operate in a cascading manner, making it challenging to rectify infeasible subgoals generated by the LLM without low-level feedback.

**Questions:**

Within Table 1, the Baguette demonstrates a volume change exceeding 30%. What does this numerical value signify, and how might it influence the manipulation process? Providing corresponding visual results would enhance clarity in understanding the observed data and its potential impact on the manipulation process.

How likely would the LLM generate unachievable subgoals? And for these situations, how would the proposed framework handle the cases?

---

> ### Author Response · Authors · 2023-11-15
> **Author response**
>
> Thanks for your valuable feedback. Below are detailed clarifications.
> > The subgoal point cloud is created using the Python code generated by the LLM, with constraints on the shape's complexity or intricacy;
>
> We appreciate your observation regarding the limitations of current LLMs in generating complex shapes. As highlighted in our discussion on limitations (Section 5), our research is not centered on advancing the capabilities of 3D generative models, but rather on proposing a novel planning framework for manipulating deformable objects. The inclusion of Point-E as an illustrative example in Figure 7 demonstrates our commitment to leveraging state-of-the-art Text-to-3D models to aid in intermediate shape generation. As LLMs continue to evolve, we anticipate more accurate and intricate shape generation that will integrate with our proposed framework.
>
> > This method necessitates users to submit comprehensive prompts, and the system's effectiveness is closely tied to the quality of the prompt data provided;
>
> We create a standardized prompt template, which simplifies user interaction to a single descriptive line. The effectiveness of this approach is further supported by our introduction of 'chain of thoughts' and 'volume preserving guidance', which are key contributors to achieving high-quality outcomes, as detailed in section 3.1. The system's dependency on prompt quality is a known factor in natural language processing applications [1,2]. The focus of this paper is not to refine the underlying language model but to spotlight our coarse-to-fine hierarchical planning framework that synergizes LLM with EMD space planning for complex manipulation tasks.
>
> [1] Wenlong Huang, Pieter Abbeel, Deepak Pathak, and Igor Mordatch. Language models as zero-shot planners: Extracting actionable knowledge for embodied agents.
>
> [2] Michael Ahn, Anthony Brohan, Noah Brown, Yevgen Chebotar, Omar Cortes, Byron David, Chelsea Finn, Chuyuan Fu, Keerthana Gopalakrishnan, Karol Hausman, et al. Do as i can, not as i say: Grounding language in robotic affordances.
>
> > The system appears to operate in a cascading manner, making it challenging to rectify infeasible subgoals generated by the LLM without low-level feedback.
>
> We concur that the LLM may not always generate perfect subgoals. Nevertheless, as shown in our updated supplementary materials (A.6), the EMD space planning algorithm's closed-loop design is robust to such variations, consistently recalibrating based on real-time observations. This adaptability is also exemplified in Figure 4, where even if initial alignment is suboptimal, subsequent steps are unaffected, validating our framework's efficacy in practical scenarios.
>
> > Within Table 1, the Baguette demonstrates a volume change exceeding 30\%. What does this numerical value signify, and how might it influence the manipulation process?
>
> This question is related to the above. The noted volume change for the Baguette in Table 1 is indicative of the algorithm's resilience to variations in object manipulation. This robustness stems from our framework treating LLM-generated plans as a directional guide rather than absolute instructions. We provide some visualizations in the updated A.5, our model can still accomplish the task because the EMD space gradient is still pointing to the correct direction.
>
> > How likely would the LLM generate unachievable subgoals? And for these situations, how would the proposed framework handle the cases?
>
> The potential for an LLM to produce unachievable subgoals is indeed a consideration in our framework. However, our empirical data indicates that the variance in LLM-generated subgoals typically remains within the bounds of what our EMD space planning algorithm can realistically achieve. The EMD space planning algorithm is adept at making incremental adjustments to align with these subgoals, which are often close approximations of the physically possible shapes.
>
> We acknowledge that our method is not a panacea for all, especially those with extreme complexity. This is a point we have transparently addressed in our discussion of the limitations of our study. Nevertheless, the novelty of our contribution rests on the synergy between LLM planning and DiffPhysics-P2P, which enables us to tackle a wider array of complex tasks than previously achievable. This represents a significant step forward, pushing the boundaries of what can be realized in the realm of deformable object manipulation.
>
> Moreover, in the instances where the LLM might propose a subgoal that is beyond the scope of current capabilities, this is not necessarily a failure but an opportunity for further research and development. Our work opens the door for advancements in the interpretation and refinement of LLM-generated plans, setting the stage for future innovations that can handle an even greater spectrum of complexity.
>
> We are happy to participate in any further discussions and we hope the responses address your concerns.

---

> > ### Comment · Reviewer_c5qh · 2023-11-22
> >
> > Thanks for the authors’ response, which clarifies the majority of my inquiries.

---

> > > ### Author Response · Authors · 2023-11-22
> > >
> > > We appreciate your acknowledgment of our response and are glad to hear it has clarified the concerns. We have strived to address all the points raised comprehensively. If you feel the revisions and clarifications have enhanced the value of our work, we hope this might be reflected in your assessment. Thank you for your thoughtful consideration.

---

### Comment · Area_Chair_uz2X · 2023-11-20
**Post-rebuttal Feedback**

Dear reviewers:

Thanks for serving as a reviewer for ICLR!

The rebuttal has been posed for a while.

Could everyone take a look and give a response?

Thanks

AC

---

### Meta-Review · Area_Chair_uz2X · 2023-12-11

**Metareview:**

This paper was reviewed by four experts in the field and received a mixed score. The main concerns are unconvincing motivation, insufficient experiments, and lack of clarity. The authors did a good job of rebuttal and addressed many of the concerns. However, the reviewers ) still feel that more work is needed to get it to the best version. The main concerns include:

1.  The motivation for using LLM for deformable manipulation is unclear.
2.  The potential of this approach for real-world deformable manipulation applications is questionable.

While the research demonstrated indeed has promise, the decision is not to recommend acceptance in its current state. The authors are encouraged to consider the reviewers' comments when revising the paper for submission elsewhere.

**Justification For Why Not Higher Score:**

1.  The motivation for using LLM for deformable manipulation is unclear.
2.  The potential of this approach for real-world deformable manipulation applications is questionable.

**Justification For Why Not Lower Score:**

NA

---

### Decision · Program_Chairs · 2024-01-16

Reject